# Mechanistic insights into coordinated *var* transcriptional switching in malaria parasites

Joseph E Visone[1,2], Francesca Florini [1,2], Evi Hadjimichael[1], Valay Patel[1] & Kirk W Deitsch [1✉]

## Abstract

**The exceptional virulence of the human malaria parasite, *Plasmodium falciparum*, is attributed to the adhesive properties of infected red blood cells and the parasite's ability to avoid antibody recognition through antigenic variation. Both properties are derived from the hypervariable surface protein PfEMP1, which is encoded by members of the multi-copy *var* gene family. Waves of parasitemia during an infection are thought to correspond to *var* transcriptional switching, enabling parasites to avoid elimination by antibodies targeting previously expressed forms of PfEMP1. The mechanisms underlying and regulating *var* transcriptional switching remain incompletely understood. Here, we show how transient activation of the *var2csa* locus mediates *var* switching, while the expression of non-coding RNAs from this locus contributes to repression of *var2csa* transcription and affects *var* switching frequencies. Furthermore, we find that an upstream open reading frame in the 5′-untranslated region of the *var2csa* transcript destabilizes the *var2csa* mRNA through the induction of the nonsense-mediated RNA decay pathway. This process promotes transcriptional activation of an alternative *var* gene. Our findings provide molecular insights into the coordinated transcriptional switching of the *var* gene family, which contributes to chronic infection.**

**Keywords** mRNA Stability; Malaria Virulence; Transcriptional Switching; Antigenic Variation; Noncoding RNA
**Subject Categories** Immunology; Microbiology, Virology & Host Pathogen Interaction; RNA Biology

## Introduction

Malaria continues to devastate subtropical and tropical regions around the globe sickening hundreds of millions of individuals annually (Venkatesan, 2025). These infections are estimated to result in up to 600,000 deaths per year, with children and pregnant women at the highest risk of disease complications. Malaria parasites belong to the genus *Plasmodium* and cause disease through replication within their host's red blood cells. Key to the pathogenesis of *P. falciparum*, the most virulent human malaria parasite species, is its ability to modify the host cell membrane with cytoadherent proteins (Miller et al, 2002). These proteins mediate sequestration within the vasculature, protecting the parasite from filtration and clearance by the spleen, but ultimately leading to many of the severe disease manifestations such as cerebral malaria and pregnancy-associated malaria. The dominant protein responsible for cytoadhesion is called *Plasmodium falciparum* erythrocyte membrane protein 1 (PfEMP1) (Hviid and Jensen, 2015).

Once PfEMP1 is surface-exposed to mediate sequestration, it is targeted by the host's adaptive immune system. Current models of antigenic variation propose that to evade antibodies that specifically recognize PfEMP1, parasites can change the isoform expressed on the red blood cell surface, altering the cells' antigenic profile. This enables parasites to avoid immune clearance and prolong the infection (Deitsch and Dzikowski, 2017). Alternative PfEMP1 isoforms are encoded by the *var* multicopy gene family (Baruch et al, 1995; Smith et al, 1995; Su et al, 1995), which consists of ~60 highly variable genes located either in subtelomeric domains or in tandem arrays within the interior regions of the chromosomes (Otto et al, 2019). Parasites typically express only a single *var* gene at a time through a process known as mutually exclusive expression (Scherf et al, 1998), although recent work indicates that this is more flexible than previously understood (Florini et al, 2025). These observations suggest that by switching expression between different *var* genes, parasites can avoid antibody recognition, resulting in the characteristic waves of parasitemia that are commonly observed in *P. falciparum* infections. Thus, by tightly controlling *var* gene switching, parasites can extend an infection for over a year (Andrade et al, 2020; Ashley and White, 2014), a time frame than can span a dry season when the mosquito vector is absent and thus ensure continued transmission.

The number of *var* genes found in the parasite's genome is relatively small when compared to other eukaryotic pathogens that employ antigenic variation, such as *Trypanosoma brucei* (Horn, 2014) and *Giardia lamblia* (Gargantini et al, 2016). Consequently, to avoid premature expenditure of this limited antigenic repertoire, not only must individual parasites limit the number of *var* genes expressed at a time, but the entire infecting population, which can number in the millions, must similarly limit cumulative expression to a small number of *var* genes at any time point in the infection (Bachmann et al, 2011). This is in stark contrast to populations of infecting *T. brucei* that have been shown to simultaneously express a heterogenous mix of variant antigens (Mugnier et al, 2015). The extremely large size of the gene family encoding the dominant surface protein in *T. brucei* and its ability to continuously generate

[1]Department of Microbiology and Immunology, Weill Cornell Medical College, New York, NY, USA. [2]These authors contributed equally: Joseph E Visone, Francesca Florini.
✉E-mail: kwd2001@med.cornell.edu

new variants provides these parasites with a virtually unlimited repertoire that cannot be exhausted. In contrast, the *var* gene family is limited to ~60 copies, thus the *var* switch rate is presumed to be slower and/or more coordinated to restrict PfEMP1 exposure within the entire infecting parasite population to a limited number of isoforms, thereby preserving the antigenic repertoire and extending the length of the infection. While the parasite does not seem to have a predetermined pattern or order to *var* gene switching, there appears to be some bias to the rate of activation and silencing of particular *var* genes (Frank et al, 2007; Horrocks et al, 2004; Recker et al, 2011). Expression is regulated at the level of transcription initiation and various histone marks and histone variants have been associated with the activation or silencing of *var* genes indicating it is an epigenetic process (Chookajorn et al, 2007; Lopez-Rubio et al, 2007; Petter et al, 2013). Genomic organization (Kraemer and Smith, 2003; Lavstsen et al, 2003), cis-acting DNA elements (Avraham et al, 2012; Deitsch et al, 2001a; Voss et al, 2006; Voss et al, 2000), and non-coding RNA transcripts (Amit-Avraham et al, 2015; Barcons-Simon et al, 2020; Diffendall et al, 2024; Epp et al, 2009; Jing et al, 2018) have all been implicated in *var* gene activation and silencing. Despite identification and characterization of these key components, a complete understanding of the mechanisms that control mutually exclusive *var* gene expression and transcriptional switching remain elusive.

With the exception of a small number of "strain-transcendent" genes (Kraemer et al, 2007; Wang et al, 2012), the *var* gene family is hypervariable, with individual parasite isolates displaying highly diverse *var* repertoires (Otto et al, 2019; Pilosof et al, 2019), thus enabling people to be infected repeatedly by different parasites with only limited cross-protective immunity. One gene, called *var2csa* (Pf3D7_1200600), displays several unique characteristics. It is universally conserved across all *P. falciparum* isolates (Otto et al, 2019), and this conservation extends to closely related species that infect chimpanzees, gorillas and bonobos (Gross et al, 2021; Otto et al, 2018). The PfEMP1 encoded by this gene, resembling an ancient form of EMP1 found in distantly related *Laverania* parasites (Otto et al, 2018), is responsible for pregnancy-associated malaria due to its binding to chondroitin sulfate A displayed by syncytiotrophoblasts in the placental vasculature (Salanti et al, 2003). Selective pressure has maintained this gene in the same chromosomal location, resistant to recombination, unlike the rest of the *var* family. Its conservation includes a unique upstream open reading frame (uORF) that regulates translation of the *var2csa* transcript into PfEMP1 (Amulic et al, 2009; Bancells and Deitsch, 2013). Besides its role in pregnancy-associated malaria, recent findings demonstrated a key secondary role for *var2csa* in the *var* switching network, acting as a "sink-node" and coordinating the switching process (Ukaegbu et al, 2015; Zhang et al, 2022). These findings led to models in which transient activation of *var2csa* throughout an infecting population of parasites can somehow unify the *var* gene family into a single coordinated network, thereby narrowing the number of genes that get activated at any time and thus conserving the repertoire. However, the molecular mechanisms underlying this process remain obscure. Moreover, how this gene can both encode a placental-specific form of PfEMP1 while also serving as a regulator of *var* gene switching has been difficult to reconcile.

Recently, we showed that *P. falciparum* can respond to environmental signals, particularly alterations in S-adenosylmethionine (SAM) levels and histone methylation potential, leading to a switch in *var* gene expression (Schneider et al, 2023). Increases in SAM levels that lead to an increase in histone methylation across heterochromatic regions of the genome result in a shift towards transcription of *var2csa*. Convergence to expression of *var2csa* over time is commonly observed in cultured wild-type parasites (Ukaegbu et al, 2015; Zhang et al, 2022), leading to the hypothesis that this gene can serve as a "default" for *var* expression (Mok et al, 2008), and providing additional evidence for its potential role in coordinating switching. However, not all parasite lines seem to respond in the same manner. When parasites are adapted to growth in culture, some laboratories observe convergence to *var2csa* expression (Andradi-Brown et al, 2024), whereas other groups do not (Lavstsen et al, 2005; Peters et al, 2007). In a similar fashion, we have observed even when parasites of the same genotype are maintained, some lines seem to readily activate *var2csa* upon stimuli and others do not. In this work, we elucidate the mechanism underlying these variations, which can be attributed to *var2csa* existing in three distinct regulatory states: one where the gene is actively transcribed and translated, one where the gene is fully silenced and a third where it undergoes transient transcription without translation, a state that facilitates transcriptional switching. The ability of *var2csa* to function as an intermediate in *var* gene switching is dependent on the conserved uORF and reinforced by nonsense-mediated mRNA decay. Moreover, we demonstrate how alternative chromatin modifications and the activation or silencing of a long non-coding RNA determine the gene's state. Our findings shed light on the complex regulatory mechanisms governing *var2csa* expression, providing crucial insights into how expression of its encoded PfEMP1 is limited to the placental environment. More broadly, our work provides a model for the central role played by the *var2csa* locus in mediating *var* gene switching and thus unifying the *var* gene family into a single coordinated network.

# Results

## Isogenic clones display differential switch rates

We recently generated an extensive collection of closely related, isogenic 3D7 subclones through limited dilution isolation (Kirkman et al, 1996; Zhang et al, 2022). These parasites lines were all derived from the same initial founding population, thereby enabling us to examine variation in *var* gene expression and switching dynamics in different parasite populations that were virtually genetically identical. Using both a standardized, *var*-specific quantitative real-time PCR (qRT-PCR) assay and whole transcriptome RNA-sequencing, we observed that despite their close genetic relationship, these parasite lines exhibited dramatically different rates and/or ability to undergo *var* gene switching. Simply by growing these parasites under standard culture conditions, some populations readily displayed easily detectable changes in *var* gene expression, becoming very heterogenous over time. We referred to these parasites as displaying a "switcher" behavior. In contrast, other parasite lines maintained a relatively homogenous *var* expression profile and did not undergo a switch from the dominantly expressed *var* gene, even after 8 months in culture. We termed such lines as "non-switchers". Representative

examples of *var* expression profiles overtime for both phenotypes are shown in Fig. 1A. The ability of these two lines to undergo *var* expression switching was profoundly different, and the *var* expression profile of the "non-switcher" line was remarkably stable when the parasites were grown under standard culture conditions.

Previous work demonstrated that *var* switching can be affected by environmental conditions, for example changes in media that affect lipid metabolism and intracellular levels of S-adenosylmethionine (SAM), the universal methyl donor for lipid synthesis and protein methylation (Schneider et al, 2023). To further explore the divergent nature of these lines' ability to undergo *var* gene switching, we performed a knockdown of the enzyme S-adenosylmethionine synthetase (SAMS), a genetic modification that we previously observed leads to the most extreme reduction in intracellular SAM levels and to a dramatic change in *var* gene expression (Schneider et al, 2023). By modifying the 3′ UTR of the locus encoding (SAMS), protein levels for this enzyme are constitutively reduced, leading to a corresponding reduction in SAM, the principal methyl donor for histone methylation. We previously showed that this results in the activation of multiple *var* genes at both the population (Schneider et al, 2023) and single cell level (Florini et al, 2025). We replicated the knockdown of SAMS in both the "switcher" and "non-switcher" backgrounds (Fig. EV1) and analyzed changes in *var* expression using qRT-PCR. In the "switcher" line, knockdown of SAMS phenocopied the previously observed results, resulting in multiple *var* genes actively transcribed within the parasite population, as expected (Fig. 1B). However, knocking down SAMS in the "non-switcher" line did not result in a switch from the active *var* gene (Fig. 1C). The remarkable stability of the "non-switcher" population was surprising given the dramatic change in *var* gene expression previously observed upon reduction in intracellular SAM levels. Taken together, these data suggest that different clonal populations can display profoundly different abilities to switch between *var* genes.

## *var2csa* expression during the asexual cell cycle varies between parasite lines

The dramatic difference in *var* gene switching propensity displayed by the "switcher" and "non-switcher" lines, despite the fact that these lines were derived from a single, clonal population of parasites, led us to investigate whether this difference might have derived from genetic changes obtained during long-term culture. Whole-genome sequencing was performed on several lines, however, no changes that correlated with *var* expression phenotypes were observed. We similarly performed RNA-Seq to determine if there were distinct changes in gene expression that might explain the difference in *var* gene switching frequency, however, comparisons of the transcriptomes identified very few differentially expressed genes that were largely limited to families of clonally variant genes encoding exported proteins (Fig. EV2 and Dataset EV1). Importantly, no changes were identified in any genes encoding predicted transcription factors or nuclear proteins, thus these two distinct phenotypes were unlikely to be due to genetic mutations or changes in the abundance of specific transcriptional regulators.

Interested in understanding the differences between these cell lines, we more deeply analyzed their *var* gene expression patterns across the full 48-h asexual cell cycle rather than simply at the peak of *var* expression at 10–20 h. We analyzed three cell lines with three different *var* expression profiles when examined at the peak of *var*

expression: a "switcher" line expressing the dominant *var* gene PF3D7_0421100, a "non-switcher" line expressing the dominant *var* gene PF3D7_0711700, and a line stably expressing *var2csa*. Parasites were tightly synchronized to a 3-h window and total RNA was extracted at different timepoints across the replication cycle (Dataset EV2). In all three lines, peak expression of the dominant *var* gene was observed in mid-ring stages between 13 and 19 h after invasion, as expected, with expression declining in the second half of the cell cycle. In the "non-switcher" line, the active *var* gene behaved as anticipated and no other *var* gene was detected throughout the cycle (Fig. 2A). Similarly, in the line expressing *var2csa*, we observed *var2csa* as the only *var* gene activated (Fig. 2B). As previously reported (Dahlback et al, 2007; Petersen et al, 2021), *var2csa* appears to activate somewhat earlier in the cycle compared to other *var* genes. In the "switcher" line however, while the active *var* gene dominated the *var* expression profile throughout the majority of the cycle, *var2csa* transcripts were surprisingly detected at the early timepoints (1–7 h post invasion) after which they rapidly declined, a phenomenon not previously observed (Fig. 2C). These transcripts were only detected at the early time point and were no longer observed later in the time course. The detection of differential, transient *var2csa* expression patterns in the "switcher" vs "non-switcher" lines reinforced our interest in investigating a potential role for this gene is regulating switching.

## Early peak of *var2csa* transcripts is associated with changes in ncRNA expression and differential chromatin assembly

The *var2csa* locus displays numerous properties that are unique within the *var* gene family, including its universal conservation in all global isolates of *P. falciparum*, its unusual promoter and upstream regulatory region (referred to as UpsE) and its propensity for transcriptional activation, leading to the suggestion that it might serve as a "default" gene for *var* expression (Mok et al, 2008). In addition, we previously identified this gene as a possible "sink node" in the *var* switching network (Ukaegbu et al, 2015; Zhang et al, 2022), highlighting the potential role of this gene in regulating expression of the entire *var* gene family. Given that we failed to detect any significant differences between the "switcher" vs "non-switcher" lines through whole-genome sequencing, we hypothesized that epigenetic alterations were likely responsible for the differences in switching frequency. We therefore sought to determine if chromatin changes were associated with the different states of *var2csa* expression. To assess chromatin modifications, we used CUT&RUN to look at histone methylation, an epigenetic mark established as a key determinant responsible for changes in *var* gene expression (Chookajorn et al, 2007; Lopez-Rubio et al, 2009). We probed for H3K9me3 and examined its occupancy in a *var2csa* expressing line as well as the "switcher" and "non-switcher" lines. As anticipated, H3K9me3 occupancy was greatly reduced at the upstream region and gene body in the *var2csa* expressing line, indicative of an open, euchromatic state. Surprisingly, however, when comparing the "switcher" and "non-switcher" lines, H3K9me3 occupancy at the promoter region of *var2csa* did not exhibit the change we hypothesized and appeared virtually identical. However, we observed a significant change in H3K9me3 occupancy within the gene body, particularly at the intron of *var2csa* (Fig. 3A).

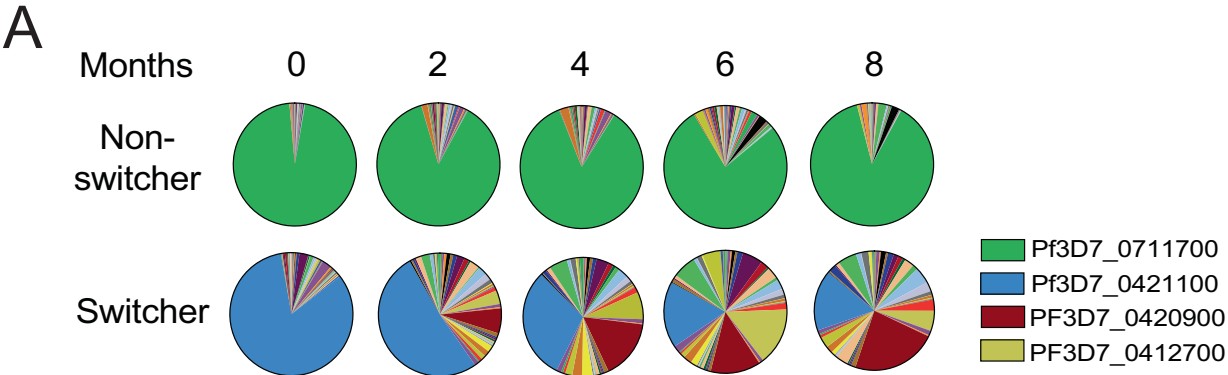

**Figure 1. Isogenic clones display different switching phenotypes.**

(A) Pie charts representing *var* expression profiles over time for a non-switcher clone (top) and a switcher clone (bottom). Expression of each gene was determined by quantitative RT-PCR and calculated as relative to seryl-tRNA synthetase (PF3D7_0717700). Each slice of the pie represents the level of expression of a different *var* gene. *var* expression was determined every two months. (B) *var* expression profiles of the switcher clone before (left) and after (right) SAMS-KD, represented as bar chart. Expression of each *var* gene was determined by quantitative RT-PCR and represented as relative to seryl-tRNA synthetase (PF3D7_0717700). *var* genes are ordered by type. (C) *var* expression profiles of the non-switcher clone before (left) and after (right) SAMS-KD, represented as bar chart. Expression of each *var* gene was determined by quantitative RT-PCR and represented as relative to seryl-tRNA synthetase (PF3D7_0717700). *var* genes are ordered by type. Source data are available online for this figure.

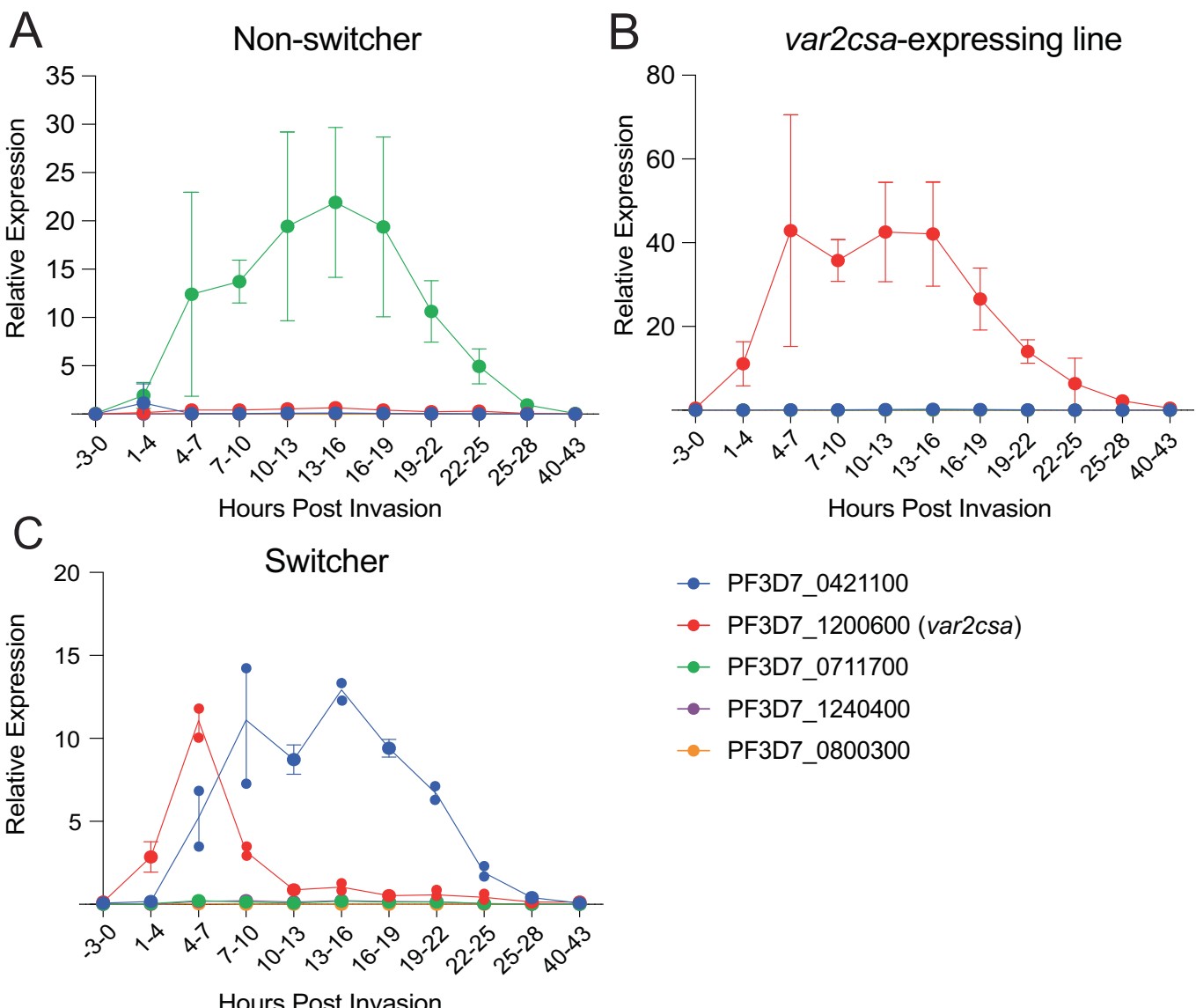

**Figure 2. *var2csa* expression over the asexual cell cycle varies between clones.**

Time course experiments showing *var* expression during the asexual cell cycle in a non-switcher clone (**A**), a *var2csa*-expressing line (**B**), and a switcher clone (**C**). Expression of each *var* gene was determined by quantitative RT-PCR and calculated as relative to seryl-tRNA synthetase (PF3D7_0717700). The three dominantly expressed *var* genes are shown along with two non-expressed *var* genes. Full *var* expression profiles for all lines at all timepoints are available in Dataset EV2 and RNAseq profiles for Pf3D7_1200600, Pf3D7_0711700 and Pf3D7_0421100 in rings and trophozoites for switchers, non-switchers and *var2csa*-expressing parasites are provided in Fig. EV5. *var* expression was determined every 3 h from tightly synchronized populations and is shown as median ± standard deviation of three independent biological replicates. For the timepoints in the switcher clone where only two replicates were available, the value of the individual data points is shown.

The intron of *var* genes is highly conserved throughout the family and contributes to regulating *var* gene expression through its promoter activity (Deitsch et al, 2001a; Dzikowski et al, 2007), although the precise mechanism is unknown. These promoters drive expression of two non-coding RNAs (ncRNAs) (Fig. 3B) (Epp et al, 2009). The first is an antisense ncRNA that has been associated with active mRNA transcription of the corresponding *var* gene (Amit-Avraham et al, 2015; Heinberg et al, 2022). The second is a sense ncRNA which has been proposed to be involved in *var* gene silencing (Epp et al, 2009). We speculated that the differential occupancy of H3K9me3 at the intron might lead to a

change in expression of this silencing ncRNA. To assess this, we designed a qRT-PCR assay using PCR primers specific to exon 2 of *var2csa* (Dataset EV3). RNA was extracted from three isogenic, recently subcloned parasite lines derived from the "switcher" and "non-switcher" populations shown in Fig. 1A. The extractions were performed on tightly synchronized, trophozoite-stage parasites when this ncRNA is known to be expressed (Kyes et al, 2003). qRT-PCR revealed that the ncRNA was significantly higher in the "non-switchers" compared to the "switchers" (Fig. 3C). To confirm this result, we performed RNA-Seq on tightly synchronized late-stage parasites. In addition to the standard protocol using amplification

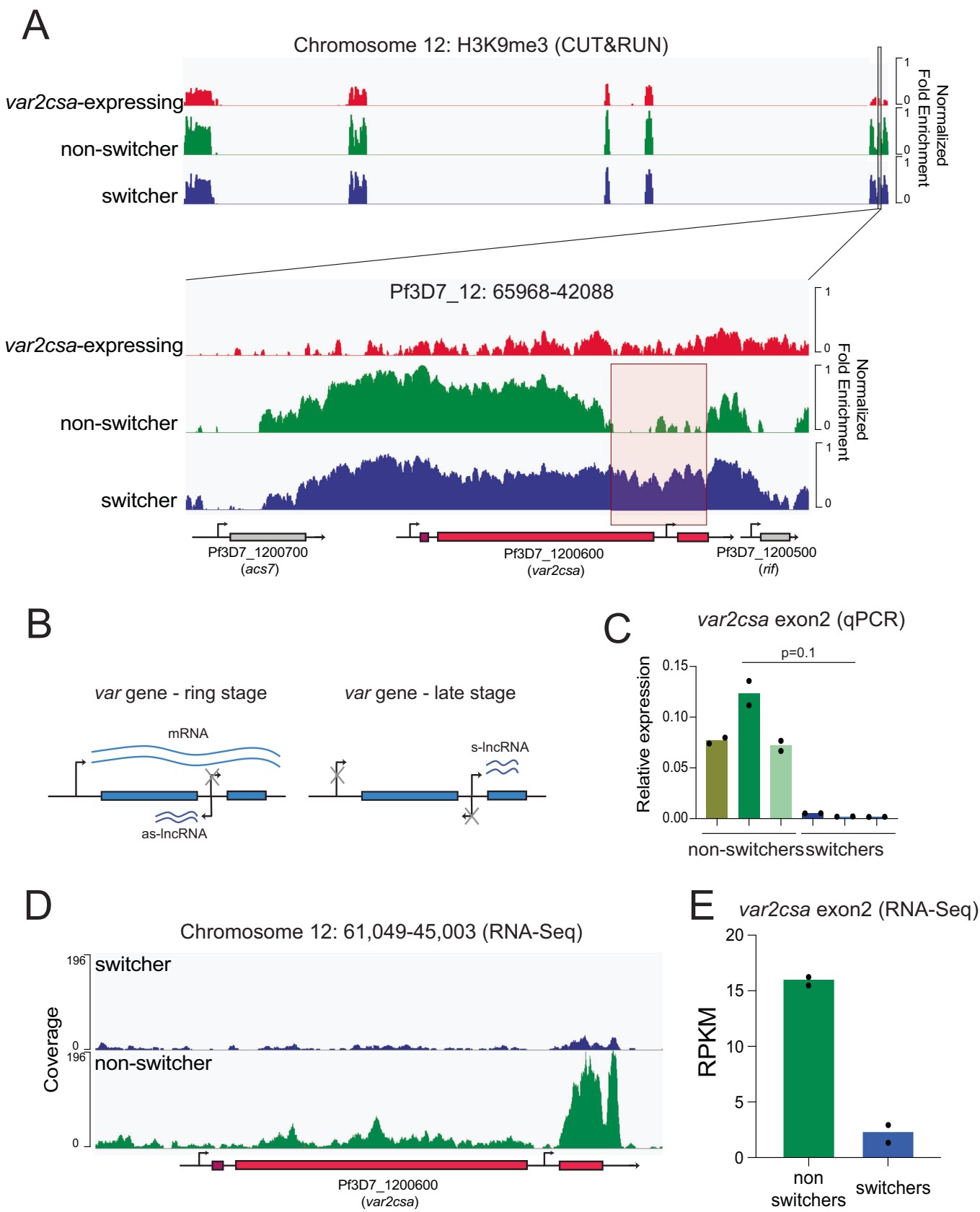

A    Chromosome 12: H3K9me3 (CUT&RUN)

*var2csa*-expressing
non-switcher
switcher

Pf3D7_12: 65968-42088

*var2csa*-expressing
non-switcher
switcher

Pf3D7_1200700 (*acs7*)    Pf3D7_1200600 (*var2csa*)    Pf3D7_1200500 (*rif*)

B

*var* gene - ring stage
mRNA
as-lncRNA

*var* gene - late stage
s-lncRNA

C    *var2csa* exon2 (qPCR)
Relative expression
p=0.1
non-switchers    switchers

D    Chromosome 12: 61,049-45,003 (RNA-Seq)
switcher
non-switcher
Coverage
Pf3D7_1200600 (*var2csa*)

E    *var2csa* exon2 (RNA-Seq)
RPKM
non switchers    switchers

**Figure 3. Differential expression of *var2csa* exon 2 nc-RNA correlates with *var2csa* mRNA expression in different lines.**

(A) Whole genome CUT&RUN was used to determine differences in H3K9me3 abundance between a *var2csa*-expressing line (red), a switcher line (blue), and a non-switcher line (green). Tracks show the relative fold enrichment versus isotype control and are scaled to noise as determined from a non-heterochromatic region. The bottom panel displays an enlarged view of the subtelomeric region of chromosome 12 that contains *var2csa* (location: 65968-42088). The genes encoded within this chromosomal region are displayed at the bottom along with their gene IDs. The arrow represents the directionality of the gene. A red box highlights the region corresponding to the *var2csa* intron and displaying the primary difference between the switcher clone and the non-switcher. (B) Figure displaying the different promoters within a *var* gene and when they are active. In ring-stage parasites, the active *var* gene transcribes a PfEMP1-encoding mRNA from the promoter upstream of the gene and a long non-coding RNA (as-lncRNA) in the antisense direction from the intron promoter. In late-stage parasites, the upstream promoter and the intron promoter in the antisense direction are silent, while the intron promoter produces a long non-coding RNA (s-lncRNA) in the sense direction. This ncRNA has been implicated in *var* gene silencing. (C) Expression of *var2csa* exon2 transcripts in three non-switcher clones (green) and three switcher clones (blue). Expression was determined by quantitative RT-PCR and plotted as relative to seryl-tRNA synthetase (PF3D7_0717700). The primers used for the assay are listed in Dataset EV3. Two technical replicates for each clone are shown. Statistical comparisons were performed on ΔCt values using a non-parametric Mann–Whitney U test (ns, $p = 0.1$). Given the small sample size, statistical power is limited. While a difference in expression is apparent, it did not reach statistical significance using a non-parametric Mann–Whitney U test. (D) RNA-Seq coverage over the *var2csa* locus on chromosome 12 for a switcher (blue) and a non-switcher (green). Reads were normalized by down-sampling. One of two biological replicates per sample is shown. (E) Quantification of reads over the lengths of *var2csa* exon 2 (chr12::47950-46750). Read count is shown as reads per kilobase per million mapped reads (RPKM) for two independent biological replicates per sample. Source data are available online for this figure.

with oligo-dT primers, we employed a cocktail of primers designed to enrich for exon 2 transcripts, which are thought to be non-polyadenylated (Epp et al, 2009) (see Methods and Dataset EV3). Importantly, any reads mapping to multiple sites in the genome were disallowed in our analysis, thus ensuring that we were precisely mapping ncRNAs to the gene from which they were transcribed. In line with the results of the CUT&RUN experiment and the qRT-PCR, we observed a much greater accumulation of *var2csa* exon 2 ncRNAs in the "non-switcher" line compared to the "switcher" (Fig. 3D,E). We suspect that the detection of very low levels of ncRNA in the "switcher" line are derived from small subpopulations of parasites that have changed to the "non-switcher" state, suggesting that parasites can transition between states, as would be expected if this is an epigenetic phenomenon. Taken together, these data are consistent with the hypothesis that the *var2csa* exon 2 ncRNA contributes to repression of *var2csa* mRNA transcription and further, that the ability to activate *var2csa* influences the parasite's ability to undergo *var* gene switching.

## In "switcher" lines, *var2csa* mRNA transcripts are unstable and get degraded through the nonsense mediated decay pathway

The identification of transient *var2csa* transcripts in the "switcher" line further implicates this locus in regulating switching events throughout the *var* gene family. We previously showed that deletion of *var2csa* led to a drastic reduction in switching frequency (Zhang et al, 2022), a phenotype similar or identical to that displayed by the "non-switchers" in which *var2csa* has been largely silenced in this population and does not display transient activation. These observations suggest that the ability to activate the *var2csa* upstream promoter is important for switching, however, why activation is transient in the "switcher" line was not clear. Additionally, why do the *var2csa* transcripts produced early in ring stages become undetectable as the cell cycle progresses and the dominant *var* transcript accumulates? We speculated that the *var2csa* uORF, known to block downstream translation of the VAR2CSA PfEMP1 (Amulic et al, 2009; Bancells and Deitsch, 2013), might be leading to changes in the stability of the *var2csa* transcript. uORFs are known to mimic premature termination codons, which can affect transcript stability by triggering degradation through the nonsense mediated decay pathway (NMD)

(Matsui et al, 2007). Further, in some systems, transcript degradation by NMD can result in regulatory feedback, called NMD-mediated transcriptional gene silencing (NMTGS)(Buhler et al, 2005; Stalder and Muhlemann, 2007), reducing or silencing the promoter of the mRNA containing the premature stop codon. If a similar mechanism exists in *Plasmodium*, it could explain why transcription of *var2csa* is unstable when the mRNA is not fully translated and provide a model for how transient *var2csa* activation could mediate switching (Fig. 4A).

To investigate whether uORF-induced NMD contributes to the instability of *var2csa* expression and is key to the "switcher" phenotype, we examined the role of the regulator of nonsense transcripts 3B (UPF3B), a protein required for NMD (He et al, 1997). Loss of UPF3B should prevent NMD-mediated degradation of the *var2csa* mRNA, even when full translation of the mRNA is prevented by the uORF, thus resulting in stabilization of the transcript (Fig. 4B). Further, if NMD-mediated degradation of the *var2csa* mRNA is required for activation of an alternative *var* gene, loss of NMD in the "switcher" line should lead to irreversible activation of *var2csa*. Thus, the "switcher" line should become "frozen" expressing *var2csa*. In contrast, since the "non-switcher" line does not express any detectable *var2csa* transcripts, we predicted that it would not be affected by the loss of NMD. To test these hypotheses, we generated a genetic disruption of the coding region of *PfUPF3B* in both "switcher" and "non-switcher" lines and confirmed the modification by whole-genome sequencing. Loss of UPF3B had no detectable effect on parasite viability and RNAseq comparisons of the knockout line to wild-type parasites detected very few changes in the overall transcriptome (Dataset EV4). However, in the "switcher" line, immediately after transfection, as soon as the genetically modified parasites were obtained, the parasites displayed exclusive *var2csa* expression at a level similar to that observed in wild-type parasites stably expressing *var2csa* (Fig. EV3), indicating the transcript had been stabilized and that these parasites were no longer able to switch away from *var2csa* expression, precisely as predicted. In contrast, this pattern was not observed in the "non-switcher" line, which continued to exclusively express the originally active *var* gene (Fig. 4C). In a second, complimentary approach, we sought to prevent recognition of the uORF by modifying the start codon from methionine to isoleucine. This single base pair modification eliminates translation of the uORF, thereby allowing unimpeded

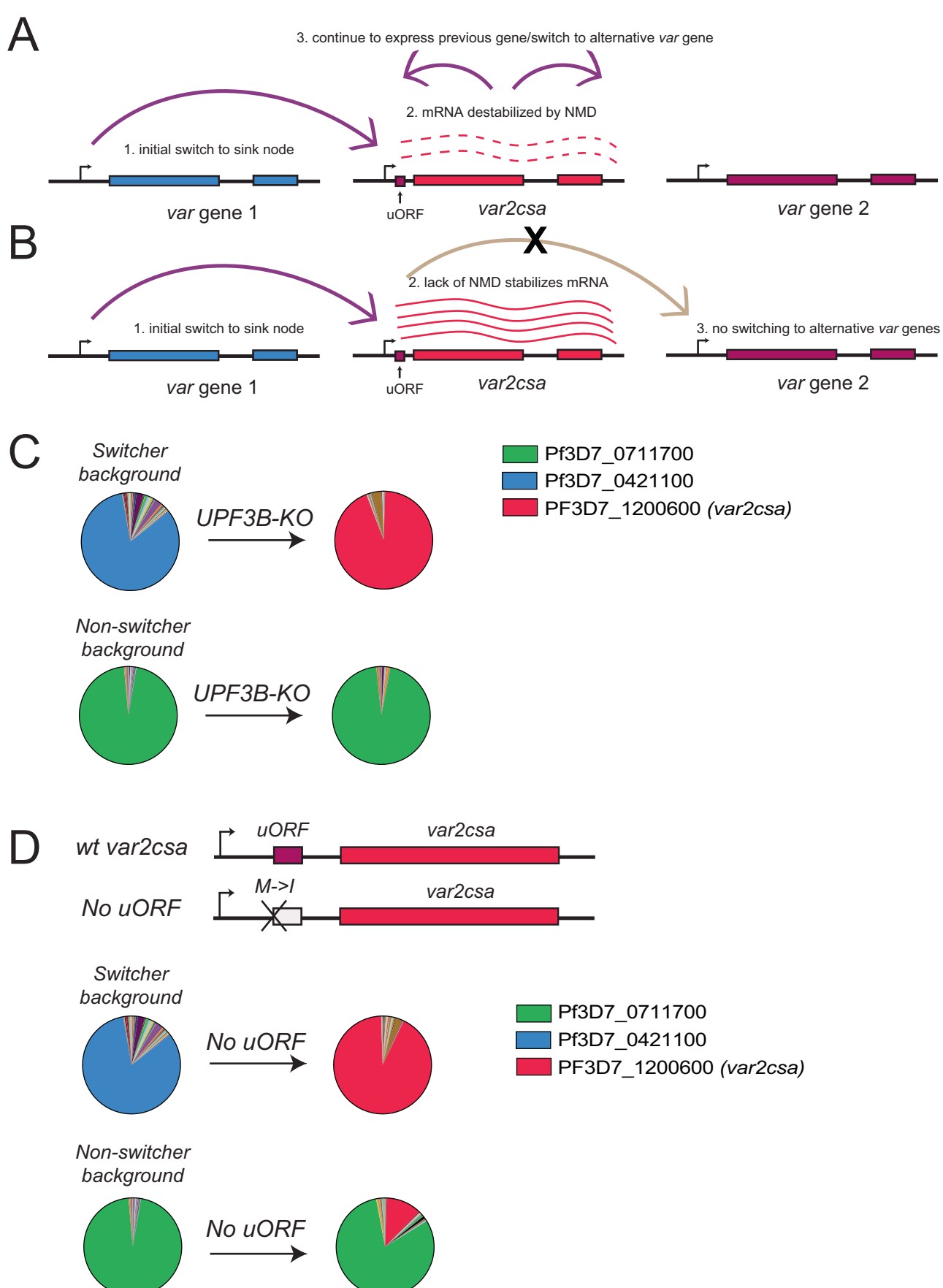

**Figure 4. The *var2csa* uORF is responsible for degradation of *var2csa* transcripts in the switcher clones.**

(A) Model for *var* gene switching and the role of *var2csa* as a sink-node. Wild-type switcher parasites transiently activate *var2csa* (red) in the beginning of the cell cycle. The uORF present upstream of the gene is recognized as a premature stop codon and *var2csa* mRNAs are destabilized through the nonsense mediated decay (NMD) pathway. This results in parasites that either continue to express the same *var* gene transcribed in the previous cell cycle (blue) or switch to a new *var* gene (purple). (B) Model for switching in NMD-deficient lines. In absence of NMD, the *var2csa* mRNA is stabilized despite the presence of the uORF, resulting in parasite unable to switch away from expressing *var2csa*. (C) Pie charts representing *var* expression profiles of a switcher clone (upper) and a non-switcher clone (bottom) before and immediately after disruption of the gene encoding UPF3B, a protein required for the NMD pathway. Expression of each *var* gene was determined by quantitative RT-PCR and represented as relative to seryl-tRNA synthetase (PF3D7_0717700). (D) Pie charts representing *var* expression profiles of a switcher clone (upper) and a non-switcher clone (bottom) before and immediately after uORF elimination. Expression of each *var* gene was determined by quantitative RT-PCR and represented as relative to seryl-tRNA synthetase (PF3D7_0717700). The schematic shows the point mutation (M > I) that eliminates the start codon and disrupts the uORF. Source data are available online for this figure.

ribosomal scanning and translation initiation of the VAR2CSA-encoding ORF (Amulic et al, 2009). In the absence of uORF translation, there is no longer a premature stop codon, and thus the mRNA is no longer a substrate for NMD. Similar to loss of UPF3B, this modification was predicted to stabilize the *var2csa* mRNA in the "switcher" line, leading to exclusive *var2csa* expression, but to have no effect on the "non-switcher" line. The modification was performed in both lines using CRISPR/Cas9-mediated gene editing with a sgRNA shielded by the modification and successful genetic modification was confirmed by Sanger sequencing (Fig. EV4). This mutation did not have any discernable effect on parasite growth, however, immediately upon modification of the locus, the "switcher" parasites displayed exclusive *var2csa* expression, phenocopying the UPF3B knockout (Fig. 4D). Once again, this was not the case for the "non-switcher", where *var2csa* remained in a silenced state in the vast majority of the parasite population, as predicted. Taken together, these data suggest that when the *var2csa* locus is not silenced by the exon2 ncRNA, *var2csa* is transiently activated. However, the presence of the uORF subjects these transcripts to recognition and degradation via the NMD pathway, which in turn leads to either continued expression of the previously active *var* gene or switching to a new *var* gene (Fig. 4A).

## Requirement for NMD to switch away from *var2csa* transcription

Current models of NMD indicate that premature stop codons within mRNAs lead to early dissociation of the ribosome from the transcript. If this occurs prior to a splice junction, the ribosome fails to displace the splice junction complex from the mRNA, leading to recruitment of the NMD complex and degradation of the transcript (Fig. 5A) (Schlautmann and Gehring, 2020; Woodward et al, 2017). We previously showed that expression of VAR2CSA requires a second round of translation initiation at the start codon of the PfEMP1 coding region (Bancells and Deitsch, 2013), which enables the ribosome to transition across the entire *var2csa* mRNA, including through the exon splice junction, thereby preventing recognition of the mRNA by the NMD complex and averting degradation. Thus, translation reinitiation is required for stabilization of the *var2csa* transcript, directly linking VAR2CSA protein translation to *var2csa* mRNA stability. However, in the UPF3B KO line, the NMD pathway is predicted to be nonfunctional, therefore the *var2csa* transcript is anticipated to be stable regardless of VAR2CSA translation. The ability to uncouple these two distinct processes provided us with an opportunity to separate their effects on *var* gene switching. For example, if the UPF3B KO parasite lines

have become "frozen" expressing *var2csa* despite not translating VAR2CSA, then NMD is key to switching away from *var2csa*.

To determine if the *var2csa* transcripts detected following disruption of UPF3B no longer required translation of the PfEMP1 encoding ORF for stabilization, we generated a line in which the efficiency of translation reinitiation was dramatically reduced. Specifically, we increased the length of the endogenous uORF by replacing it with an ORF encoding mNeonGreen (mNG) (Fig. 5B2). The mNG coding length is 711 bp, which is approximately double the length of the endogenous uORF of 360 bp. Previous work indicated that increasing the length of the uORF by this amount greatly diminishes translation reinitiation, nearly eliminating translation of the downstream ORF (Amulic et al, 2009; Bancells and Deitsch, 2013). We observed that compared to wild-type parasites, the *var2csa* mNG-uORF parasites did not converge to *var2csa* after long-term culture (Fig. 5C), suggesting that these parasites might be unable to stabilize the *var2csa* mRNA. To determine if translation of the downstream reading frame is possible with the longer mNG-uORF, we inserted a Blasticidin-S-Deaminase selectable marker in place of the *var2csa* coding region in both wild-type parasites and parasites with the mNG-uORF-modified locus, thus enabling us to select for translation reinitiation (Fig. 5B/3,4). Upon selection with blasticidin, the *var2csa-BSD* parasites recovered after approximately 10 days and *var2csa* was the dominant transcript (Fig. 5D,E), demonstrating selection of parasites that were efficiently reinitiating translation at the BSD start codon. In contrast, we were unable to recover *var2csa* mNG-uORF BSD parasites following blasticidin selection, indicating that the mNG-uORF parasites cannot overcome the translational block imposed by the longer mNG-uORF (Fig. 5D). To verify the role of NMD in destabilizing *var2csa* transcripts that are not fully translated, we generated a mNG-uORF line lacking UPF3B. Remarkably, this line displayed stable *var2csa* transcription immediately upon recovery of the genetically modified parasites, even in the absence of blasticidin selection (Fig. 5F), similar to the immediate convergence to *var2csa* previously observed when UPF3B was knocked out in a wild-type "switcher" line. Thus, despite an inability to translate the downstream ORF, in the absence of NMD these parasites stably expressed a full-length *var2csa* transcript. As expected, the mNG ORF was translated in these lines, as indicated by the green fluorescence detected in the parasites (Fig. 5G). These data further support the role of the uORF and NMD in the control of *var2csa* transcript stability. Further, as inferred from Fig. 5F, when the *var2csa* transcript cannot be degraded, parasites appear to be unable to switch away from transcribing the *var2csa* locus.

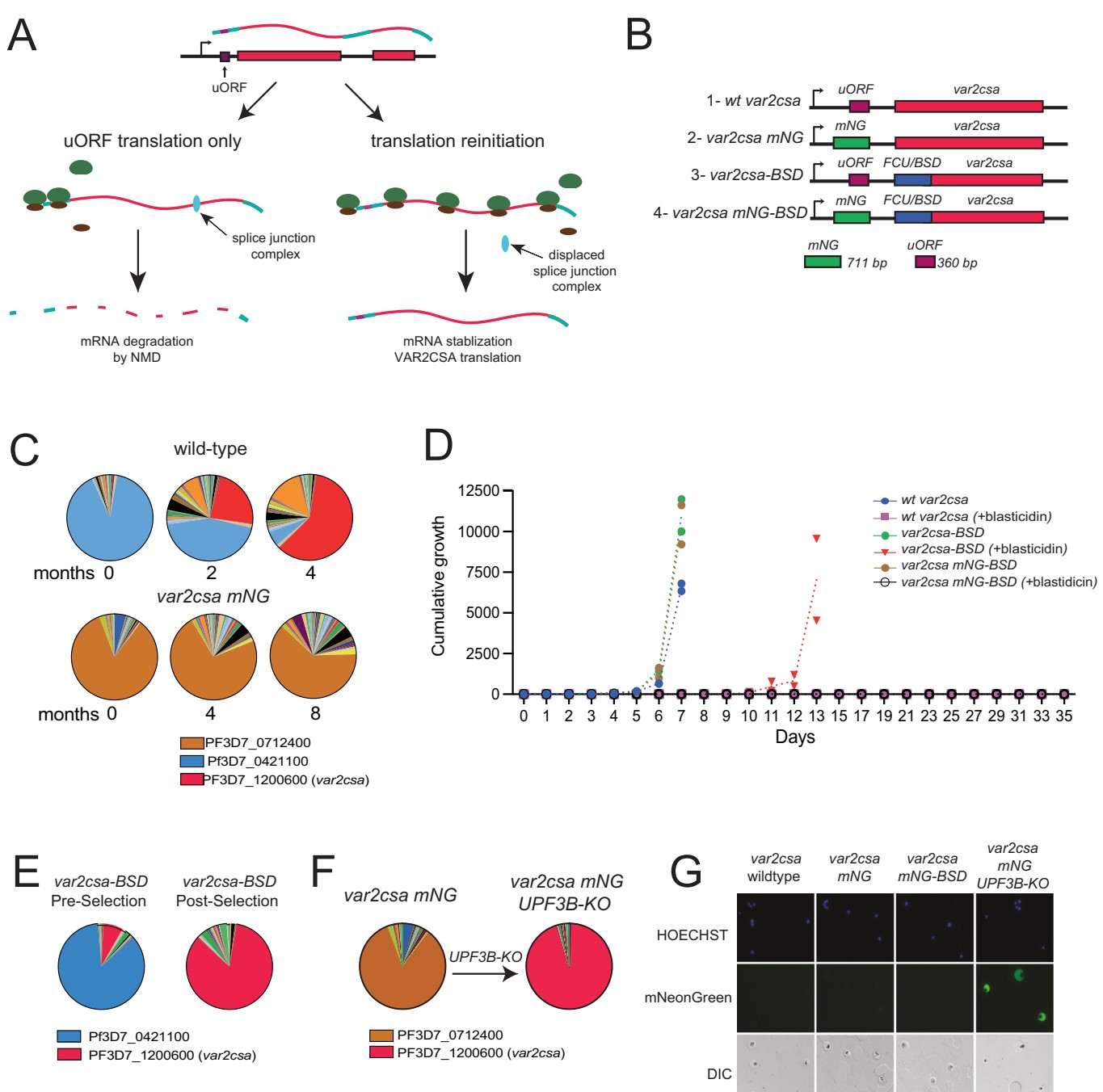

## VAR2CSA translation results in *var2csa* transcript stabilization

As demonstrated by the abundance of *var2csa* transcripts in pregnancy-associated malaria (Duffy et al, 2006), the frequent observance of *var2csa* transcripts stably expressed by cultured parasites (Mok et al, 2008) and by previous work with experimental constructs (Bancells and Deitsch, 2013), wild-type parasites are fully capable of stabilizing *var2csa* mRNAs. Given the work described above, we hypothesized that this requires translation through the uORF, reinitiation at the start codon of the PfEMP1

encoding ORF, and continued translation through the remainder of the mRNA, including the exon splice junction (Fig. 5A). Previous work indicated a requirement for Plasmodium Translation Enhancing Factor (PTEF) (PF3D7_0202400) to overcome the translation block imposed by the uORF (Chan et al, 2017). To assess if PTEF expression correlated with *var2csa* stable transcription in our wild-type lines, we analyzed several clonal lines by qRT-PCR (Dataset EV5). When comparing clones expressing *var2csa* to non-*var2csa* expressing clones, we always observed high levels of PTEF expression in parasite populations stably expressing *var2csa* (Fig. 6A). To further validate VAR2CSA translation, we

**Figure 5.** *var2csa* transcription and translation are uncoupled upon NMD disruption.

(A) Model for *var2csa* transcription and translation when only the uORF is translated (left), resulting in mRNA degradation by NMD, or in the case of translation reinitiation and translation of the VAR2CSA protein (right). Translation reinitiation results in transition of the ribosome across the full length of the mRNA, displacing the splice junction complex and stabilizing the transcript. (B) Schematic of the different modifications to the *var2csa* locus performed by CRISPR/Cas9-mediated genome editing. The *var2csa* uORF is shown in purple, mNeonGreen (mNG) in green, the *var2csa* coding region in red and the selectable markers cassette Blasticidin-S-Deaminase (BSD)/ yFCU (Braks et al, 2006) in blue. (C) Pie charts representing *var* expression profiles over time for a wild-type line (top) and upon substitution of the *var2csa* uORF with mNG (bottom). Expression of each gene was determined by quantitative RT-PCR and calculated as relative to seryl-tRNA synthetase (PF3D7_0717700). Each slice of the pie represents the level of expression of a different *var* gene. *var* expression was determined every two months. (D) Growth over time displayed as cumulative parasitemia of wild-type and modified lines, with or without blasticidin selection. Parasitemia was measured daily by flow cytometry and thin smears, and results are displayed as two biological replicates. (E) Pie charts representing *var* expression profiles of the *var2csa-BSD* line before (left) and after (right) selection with blasticidin. Expression of each gene was determined by quantitative RT-PCR and calculated as relative to seryl-tRNA synthetase (PF3D7_0717700). Each slice of the pie represents the level of expression of a different *var* gene. (F) Pie charts representing *var* expression profiles of the *var2csa mNG* line before (left) and after (right) deletion of UPF3B. Expression of each gene was determined by quantitative RT-PCR and calculated as relative to seryl-tRNA synthetase (PF3D7_0717700). Each slice of the pie represents the level of expression of a different *var* gene. (G) Live-cell fluorescence microscopy images for the different lines stained with the DNA dye Hoechst 33342 (blue). DIC = differential interference contrast. Scale bar: 10 μM. Source data are available online for this figure.

investigated if *var2csa* expressing parasites were capable of expressing VAR2CSA PfEMP1 on the erythrocyte surface. Specifically, we determined if *var2csa* expressing parasites were able to bind to purified CSA. Indeed, compared to non-*var2csa* expressing controls, *var2csa* expressing parasites showed significantly higher binding to CSA (Fig. 6B). With respect to the non-*var2csa* expressing clones, both switcher and non-switcher lines were included and neither displayed any detectable binding, indicating that the early peak of *var2csa* mRNA observed in a switcher line (clone 1, Dataset EV5) does not result in any detected binding to CSA. In a parallel approach, we examined surface export of VAR2CSA using an anti-VAR2CSA antibody (PAM4.1, (Barfod et al, 2010)) with an anti-human FITC secondary antibody. Comparing the *var2csa* expressing lines to non-*var2csa* expressing lines, we observed a clear shift in FITC signal indicating that parasites stably expressing *var2csa* mRNA are efficiently translating and exporting VAR2CSA (Fig. 6C,D). These experiments are consistent with the results obtained using transgenic parasite lines and indicate that stable expression of *var2csa* mRNA occurs when the PfEMP1-encoding ORF is translated. Taken together, the data presented here highlight how the *var2csa* locus plays a key role in two distinct processes, coordinating *var* gene expression switching and placental binding, providing an explanation for its universal conservation throughout the *P. falciparum*, *P. reichenowi*, *P. praefalciparum* and *P. lomamiensis* lineages (Gross et al, 2021).

## Discussion

The experiments described above provide a mechanistic model for how the highly conserved *var2csa* locus can function to coordinate *var* gene switching and antigenic variation in *P. falciparum*. The model incorporates three expression states of *var2csa* and is summarized in Fig. 7. Expression of ncRNAs initiating from the intron is associated with a silenced locus, resulting in a *var2csa* upstream promoter that is fully repressed and not compete with the previously expressed *var* gene, thereby suppressing transcriptional switching (Fig. 7, panel A). Our detection of high levels of ncRNA expression from the *var2csa* locus in the "non-switcher" lines (Fig. 3) is consistent with this model. In contrast, the "switcher" lines displayed little detectable ncRNA expression and the *var2csa* promoter appears to be active early in the cell cycle, as shown by detection of *var2csa* transcripts between 2-5 h after invasion

(Fig. 2). An active *var2csa* promoter can compete with the previously expressed *var* gene, thereby initiating *var* gene switching (Fig. 7, panel B). The presence of the uORF in the *var2csa* transcript prevents translation of the PfEMP1 encoding open reading frame, leading to degradation of the mRNA by NMD and silencing of the locus by Nonsense Mediated Transcriptional Gene Silencing (NMTGS) (Buhler et al, 2005; Stalder and Muhlemann, 2007), resulting in switching to an alternative *var* gene. Importantly, this model predicts that disruption of NMD or mutation of the uORF start codon will result in parasites being unable to complete the switch and being "frozen" expressing *var2csa*, precisely what we observed (Fig. 5). This model also predicts that the promoter of the gene that becomes activated is similarly in an open chromatin state, enabling it to compete with *var2csa* and the previously active gene. A recent analysis of *var* gene expression at the single cell level detected individual parasites expressing multiple *var* genes simultaneously (Florini et al, 2025), consistent with multiple *var* gene promoters competing for activation. Lastly, parasites can reinitiate a second round of translation at the start codon of the PfEMP1 ORF of the *var2csa* mRNA, thereby stabilizing the transcript and resulting in stable VAR2CSA expression (Fig. 7, panel C), as we observed in Fig. 6. Reinitiation of translation could result from exposure to the placental environment, similar to how environmental signals alter uORF-mediated translational repression in model systems (Renz et al, 2020; Young and Wek, 2016), which would ensure placenta-specific expression of VAR2CSA. Together, these three hypothetical states of *var2csa* provide an explanation for the "switcher", "non-switcher" and "*var2csa* expressor" phenotypes we observe in parasite cultures. Importantly, this model is focused primarily on the epigenetic regulation of the promoters that control *var* mRNA expression and does not incorporate possible molecular functions of the *var2csa* transcript itself, the intron-driven ncRNAs from the rest of the *var* gene family, or other ncRNAs previously implicated in regulating *var* gene expression (Barcons-Simon et al, 2020; Diffendall et al, 2024; Fan et al, 2020). Thus, a full understanding of this complicated regulatory pathway will require additional work.

The disruption of the NMD pathway resulted in stabilization of the *var2csa* transcript in the "switcher" lines, suggesting that NMD directly degrades the *var2csa* transcript when it is not fully translated. This conclusion is consistent with similar stabilization of the mRNA when the start codon of the uORF is mutated. While it is possible that loss of NMD has other effects on the overall biology of the parasite

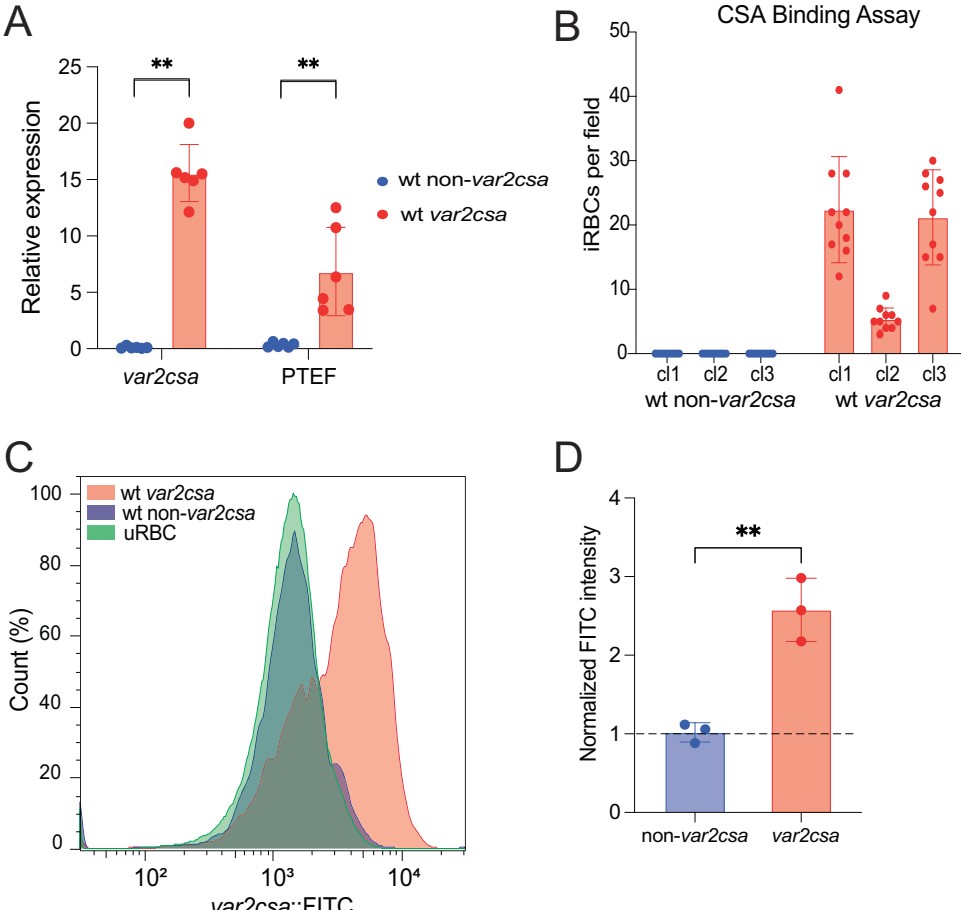

**Figure 6. VAR2CSA translation is associated with *var2csa* transcript stabilization.**

(A) Expression of *var2csa* (left) and Plasmodium Translation Enhancing Factor (PTEF, right) in six wild-type 3D7 clones expressing *var2csa* (red) and six wild-type 3D7 clones expressing a *var* gene other than *var2csa* (blue). The full *var* profiles are provided in Dataset EV5. Expression was determined by quantitative RT-PCR and calculated as relative to seryl-tRNA synthetase (PF3D7_0717700). Bar shows mean ± standard deviation of the six clones, while dots show individual mean of two technical replicates per clone. Statistical comparisons were performed on ΔCt values using a non-parametric Mann–Whitney U test (**$p = 0.0022$). (B) Number of infected RBCs (iRBCs) per independent field ($n = 10$) bound to CSA-coated plates in three NF54 wild-type clones expressing a *var* gene other than *var2csa* (blue) and three NF54 wild-type clones expressing *var2csa* (red). Bar shows mean ± standard deviation. (C) Example of flow-cytometry with PAM1.4 IgG for a wild-type NF54 clone expressing a *var* gene other than *var2csa* (blue, gated infected RBC), a wild-type NF54 clone expressing *var2csa* (red, gated infected RBC) and uninfected RBCs (green). Histogram shows normalized cell count over FITC intensity. (D) Quantification of mean FITC intensity for six NF54 clones, three expressing *var2csa* (red) and three expressing a gene other than *var2csa* (blue). The *var* expression panels for all the clones are included in Dataset EV5. As FITC intensity of infected RBCs is normalized to FITC intensity of uRBCs, the dotted line indicates no difference between uRBCs and iRBCs. Bar shows mean ± standard deviation. An unpaired t-test indicates a **$p = 0.003$. Source data are available online for this figure.

which indirectly result in stable *var2csa* expression, it is noteworthy that we could detect no effect on parasite replication upon loss of NMD and little to no effect on the transcriptome as detected by RNA-seq (Dataset EV4). Similar conclusions were described by McHugh and colleagues who disrupted NMD by knocking out UPF2, although they reported a small increase in the detection of intron retention (McHugh et al, 2023). It is therefore possible that the primary function of NMD is in regulating *var* gene expression in asexual parasites, while this pathway might have additional roles in other stages of the parasite lifecycle when regulation of mRNA stability is prominent, for example during the gamete to ookinete transition (Mair et al, 2006; Mair et al, 2010).

Our experiments revealed a close association between expression of the intron-driven ncRNA, complete silencing of the *var2csa*

locus and the "non-switcher" phenotype (Fig. 3). We therefore propose that this non-coding RNA plays a role in fully silencing the *var2csa* upstream promoter and thereby suppresses overall switching (Fig. 7). This suppressed switching is similar to what we observed when the *var2csa* locus was deleted from the genome (Zhang et al, 2022), providing additional support for transient activation of *var2csa* in initiating switching. A previous study from Bryant and colleagues (Bryant et al, 2017) explored the potential role of the *var2csa* intron by using genome editing to delete it from the locus. They observed increased rates of *var* gene switching and upregulation of *var2csa* expression, similar to what we observed in our "switcher" lines when the intron promoter is silent and does not express the ncRNA. Thus, deletion of the intron appears to phenocopy parasite lines in which the intron promoter is

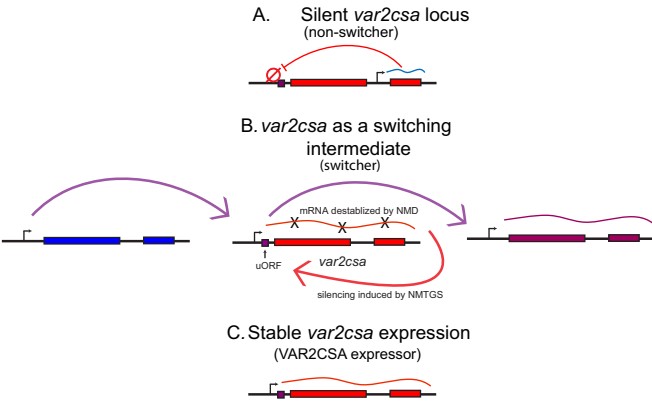

**Figure 7. *var2csa* can exist in three different states.**

Model for coordinated *var* gene switching in relation to *var2csa* state. (A) When the *var2csa* locus is actively transcribing a ncRNA, the upstream promoter is in heterochromatin and overall switching is strongly repressed. This is consistent with the "non-switcher" phenotype. (B) In the absence of ncRNA expression, the *var2csa* upstream promoter can compete with the promoter of the active *var* gene, initiating a potential switching event. This leads to the "switcher" phenotype. When switching is initiated in the absence of a placenta, *var2csa* functions as a switching intermediate or "sink node". In this state, NMD degrades the mRNA and transcription switches away from *var2csa*, either back to the same *var* gene expressed in the previous cell cycle or to an alternative *var* gene. (C) In the presence of a placenta, translation of the *var2csa* mRNA is stabilized, NMD is not activated, and parasites maintain expression of the *var2csa* locus.

transcriptionally silent. The concordance of the results of this previous study and the results reported here provide additional support for the model shown in Fig. 7. The role of the intron-driven ncRNAs might extend to the rest of the *var* gene family. For example, Bachmann and colleagues recently reported exceptionally high activation rates of a specific type C *var* gene in the 7G8 parasite line (Wichers-Misterek et al, 2023), even after transmission through the mosquito. Examination of this locus (Pf7G8_040025600) indicates a substantial deletion within the intron, including the portion that includes the ncRNA promoter (Epp et al, 2009), providing a likely explanation for the high rate of activation of this gene. The conserved sequence and structure of *var* introns extends throughout the entire Laverania clade and thus predates the evolution of the UpsA/B/C promoter types and *var2csa* (Gross et al, 2021), further implicating an important role for this element.

The model shown in Fig. 7 provides insights into how the unusual characteristics of the *var2csa* locus could enable it to function as a "sink node" or "switching hub" that coordinates *var* gene switching. It incorporates many concepts previously suggested as potentially playing a role in *var* gene regulation, including promoter competition (Dzikowski and Deitsch, 2008; Fastman et al, 2012; Voss et al, 2006) and a role for ncRNAs (Epp et al, 2009; Heinberg et al, 2022). It also provides an explanation for previous observations that *var2csa* appears to be a default gene in cultured parasites (Mok et al, 2008; Ukaegbu et al, 2015) and examples of *var2csa* transcripts detected in non-pregnant individuals (Bachmann et al, 2019; Duffy et al, 2006; Lavstsen et al, 2005). While *var2csa* transcripts are very frequently detected in samples obtained from non-pregnant individuals (Chaikitgosiyakul et al, 2025;

Rovira-Vallbona et al, 2011), the level of expression is much lower than observed in samples obtained during pregnancy (Duffy et al, 2006; Tuikue Ndam et al, 2005), consistent with the detection of the early peak of *var2csa* transcripts in "switcher" parasites and thereby suggesting that "switchers" are very common in natural infections. However, many questions remain. For example, if the role of *var2csa* as a switching intermediate coordinates which *var* gene is activated within large populations of parasites, switching away from *var2csa* must be biased to a single or small number of genes, as was hypothesized previously by Recker and colleagues (Recker et al, 2011). We and others have similarly observed biased activation of specific *var* genes (Enderes et al, 2011; Florini et al, 2025; Noble et al, 2013), however, the mechanism underlying this phenomenon is not known. Our observation that intron-driven ncRNA expression influences the frequency *var2csa* activation suggests that a similar phenomenon might apply to the rest of the *var* gene family. If so, this could enforce an activation hierarchy on the family, with genes that do not express the ncRNA being preferentially activated. It is also not known how parasites transition between the different *var2csa* expression states. In our experiments, the "switcher", "non-switcher" and "*var2csa* expressor" lines were all derived from a single, clonal population of parasites, thus parasites can clearly transition between these states. We also observe very low rates of transition between our "switcher" and "non-switcher" phenotypes, for example in Fig. 4D in which a small proportion of the "non-switcher" population displays *var2csa* expression after mutation of the uORF start codon, indicating these parasites have activated the *var2csa* promoter. Whether different parasites strains with different genetic backgrounds display varying rates of "switcher" and "non-switcher" frequencies is not yet known, nor is the potential influence of *var* promoter types (UspA, B or C) on switching phenotype understood. If promoter competition is a key attribute of switching, as we hypothesize, this is likely. Lastly, environmental changes have been shown to influence stable expression of *var2csa* (Schneider et al, 2023), and a similar response was reported for activation of *ap2-g* and sexual differentiation (Brancucci et al, 2017; Harris et al, 2023). In these previous studies, changes in environmental conditions were shown to influence the epigenetic state of mRNA promoters, however, a similar concept could apply to the transcriptional state of the *var2csa* intron promoter, thus potentially changing a parasite between switcher states. A more complete understanding of the environmental conditions encountered by parasites over the course of natural infections will be required to fully understand the nuances of *var* gene expression switching.

## Methods

**Reagents and tools table**

| Reagent/Resource | Reference or Source | Identifier or Catalog Number |
|---|---|---|
| **Experimental models** | | |
| *P. falciparum* (strain 3D7) | BEI Resources | MRA-1001 |
| *P. falciparum* (strain NF54) | BEI Resources | MRA-1000 |
| **Recombinant DNA** | | |
| PfSAMS-KD | Harris et al, 2023 | PMID: 37277533 |

| Reagent/Resource | Reference or Source | Identifier or Catalog Number |
|---|---|---|
| UPF3B-KO | This study | |
| uORF-M1I | This study | |
| var2csa mNG | This study | |
| var2csa BSD | This study | |
| pUF1_Cas9 | Ghorbal et al, 2014 | PMID: 24880488 |
| **Antibodies** | | |
| H3K9me3 antibody ChIP Grade (polyclonal) Rabbit IgG | Abcam | ab8898 |
| PAM1.4 IgG | Barfod et al, 2010 | PMID: 21078904 |
| Anti-human IgG FITC | Sigma-Aldrich | I2136 |
| Anti-Hsp70 | StressMarq | SPC-186 |
| Anti-PfSAMS | Schneider et al, 2023 | PMID: 37068249 |
| Anti-Rabbit IgG | Agilent Dako | P0399 |
| **Oligonucleotides and other sequence-based reagents** | | |
| qPCR primers | This study | Dataset EV2 |
| var qPCR primer | Salanti et al, 2003 | PMID: 12823820 |
| Exon2-specific enrichment primers | This study | Dataset EV2 |
| **Chemicals, Enzymes and other reagents** | | |
| AlbuMAX II Lipid-Rich BSA | Thermofisher | 1102037 |
| RPMI 1640 | Corning | 45000-604 |
| WR99210 | Jacobus Pharmaceuticals | |
| Blasticidin S | Gibco | R210-01 |
| DSM1 | Sigma-Aldrich | 533304001 |
| TRIzol | Thermofisher | 10296028 |
| DNase I | Thermofisher | 18068015 |
| SuperScript II Reverse Transcriptase | Thermofisher | 18064014 |
| iTaq Universal SYBR Green Supermix | Biorad | 1725124 |
| Chondroitin Sulfate A | Sigma-Aldrich | C9819 |
| **Software** | | |
| SpectroFlo | Cytek Biosciences | v3.3.0 |
| FlowJo | FlowJo LLC | v10 |
| R | R Foundation | v4.1.2 |
| GraphPad | Prism | v10.1.1 |
| **Other** | | |
| Illumina DNA Prep Kit | Illumina | 20091654 |

## Parasites culture

Parasites (*P. falciparum* strain 3D7/NF54) were maintained following standard procedures at 5% hematocrit in RPMI 1640 medium supplemented with 0.5% Albumax II (Invitrogen) in an atmosphere containing 5% oxygen, 5% carbon dioxide, and 90% nitrogen at 37 °C. For growth curves, parasites were synchronized at ring-stage with 5% sorbitol and diluted to an initial parasitemia of 1% in a total culture volume of 5 ml. Parasitemia was calculated daily on an Aurora flow cytometer with SpectroFlo (Cytek Biosciences) and verified by thin smear with Giemsa. Changes in parasitemia were calculated by multiplying the daily parasitemia by the exponential dilution factor.

## Generation of transgenic lines

Parasites were transfected by electroporation using DNA-loaded RBCs as previously described (Deitsch et al, 2001b). Transfected parasites were obtained using selection by WR99210, Blasticidin S or DSM1, and clonal lines were obtained by limited dilution (Kirkman et al, 1996). The PfSAMS knockdown replicate lines were obtained as previously described (Harris et al, 2023). These lines display reduced PfSAMS expression and constitutively altered *var* gene expression, even in the absence of glucosamine (Schneider et al, 2023), therefore no glucosamine was added to the cultures. Specific modifications to the *var2csa* and *UPF3B* loci were performed via CRISPR-targeted plasmid integration using derivatives of the plasmids pL6_eGFP and pUF1_Cas9 for CRISPR/Cas9-based genome editing as described (Ghorbal et al, 2014). Homology blocks were PCR amplified from 3D7 genomic DNA using specific primers (see Dataset EV3 for primers list) and inserted by IN-Fusion cloning (Clontech, Takara Bio USA, Mountain View, CA, USA). Plasmid integration was validated by PCR across the site of integration and whole-genome sequencing.

## RNA extraction, cDNA synthesis and qRT-PCR

For *var* panels displaying qRT-PCR data, RNA was extracted from ring-stage parasites 48 h after synchronization with 5% Sorbitol. For qRT-PCR analysis of exon2 transcripts in Fig. 3C, RNA was extracted from late-trophozoites 24 h after synchronization with additional synchronization using Percoll/Sorbitol column isolation (Aley et al, 1984; Rivadeneira et al, 1983). RNA was extracted with TRIzol (Invitrogen) and purified on PureLink (Invitrogen) columns following the manufacturer's protocols. To eliminate genomic DNA, RNA was treated with DNase I (Invitrogen). cDNA was synthesized from 1 μg of RNA with random hexamers with SuperScript II Reverse Transcriptase (Invitrogen), according to manufacturer instructions. Established sets of *var* primers (Salanti et al, 2003) were employed to determine *var* transcription through qRT-PCR. This primer set was experimentally validated for equal amplification efficiency (Salanti et al, 2003). All reactions were performed in 10 μl volumes in 384-well plates using iTaq Universal SYBR Green Supermix (Bio-Rad) in a QuantStudio 6 Flex (ThermoFisher). Each reaction consisted of 5 μl of SYBR Green master mix, 2 μl of a forward and reverse primer mixture, and 3 μl of 1:10 diluted cDNA. The reactions were set up in 384-well plates, sealed with a plate sealer, and centrifuged for 30 s using a plate centrifuge. The qPCR conditions were as follows: (1) 50 °C for 2 min; (2) 95 °C for 3 min; (3) 95 °C for 15 s; (4) 54 °C for 40 s; (5) 60 °C for 1 min. Steps 3–5 were repeated for 40 cycles. $\Delta$CT for each primer pair was determined by subtracting the individual CT value from the CT value of *seryl-tRNA synthetase* (PF3D7_0717700/PfIT_020011400) and converting to relative copy numbers with the formula $2^{\Delta CT}$. Relative copy numbers are plotted using GraphPad Prism 10 or Microsoft Excel as bar graphs and pie charts. All additional primers are listed in Dataset EV3.

## Time-course experiments

For the experiments in Fig. 2, ring-stage parasite were initially synchronized with 5% sorbitol. 24 h after sorbitol synchronization, late-stage parasites were isolated using percoll/sorbitol gradient centrifugation (Ginsburg et al, 1987) and allowed to reinvade for 3 h while shaking continuously. 0–3 h ring-stage parasites were isolated through a second percoll/sorbitol gradient centrifugation followed by sorbitol. RNA was extracted as described above every 3 h.

## RNA sequencing

Parasites were prepared for RNA-Sequencing with double percoll/sorbitol synchronization as described above. RNA was isolated at 16–19 h post-invasion for ring-stage and at 32–35 h post-invasion for late-stage. RNA was extracted with TRIzol (Invitrogen) and purified on PureLink (Invitrogen) columns, including an on-column DNase I treatment, following the manufacturer's protocols. RNA integrity was assessed using Tapestation (Agilent). For ring-stage, cDNA was prepared using oligo-dT as a primer for reverse transcription. For late-stage, cDNA was prepared using a custom mixture of oligo-dT and exon2-specific enrichment primers (see Dataset EV3). Alignment of Exon 2 sequences enabled the identification of the most highly conserved region shared between all *var* genes. For genes with non-perfect matches, an additional oligo was added to the pool resulting in 31 exon 2-specific oligos. cDNA libraries were not strand-specific. Libraries were prepared using Illumina DNA Prep Kit and sequenced using an Illumina NovaSeq X Plus sequencer with paired-end 150 protocol performed by the Genomics Core Laboratory at Weill Cornell Medicine. Quality of the reads was assessed using MultiQC (version 1.14) (Ewels et al, 2016). Reads were processed using trimmomatic (version 0.39) (Bolger et al, 2014) and mapped to the *P. falciparum* 3D7 reference genome (release 68) (Warrenfeltz et al, 2018) using STAR (version 2.7.10b) with the STAR argument --star-args = '--outFilterMultimapNmax 1' to prevent multimapping of *var* gene reads due to high similarity (Dobin et al, 2013), thus only uniquely mapping reads were allowed. Differential expressional analysis was performed using DESeq2 (v.1.36) with a false discovery rate cutoff of <0.05. This analysis was conducted with R (v.4.1.2) in Rstudio (2022.2.2.485) (RStudio Team (2022). RStudio: Integrated Development Environment for R. RStudio, PBC, Boston, MA, URL http://www.rstudio.com/). Total reads for individual samples ranged from 115 million to 48 million. For visualization, reads were normalized by random downsampling to 48 million using samtools (samtools 1.16.1) (Li et al, 2009) and visualized using IGV (version 11.0.13) (Robinson et al, 2011).

## Whole-genome sequencing

For whole-genome sequencing, genomic DNA was extracted from late-stage parasites using phenol-chloroform extraction followed by ethanol precipitation. To eliminate RNA contaminants, DNAs were treated with RNase cocktail (Invitrogen) and ethanol precipitated once again. DNA concentrations were determined using Qubit dsDNA HS assay (Thermo Fisher Scientific) and libraries were prepared with an Illumina DNA Prep kit. Sequencing was performed on the NovaSeq X Plus sequencer with paired-end

protocol performed by the Genomics Core Laboratory at Weill Cornell Medicine. Reads were trimmed using trimmomatic (version 0.39). Genome indexing and alignment were performed with bwa aligner (version 0.7.17) (Li and Durbin, 2009). Sequencing duplicates were removed using picard (v2.27.1) ("Picard Toolkit." 2019. Broad Institute, GitHub Repository. https://broadinstitute.github.io/picard/; Broad Institute). Sequences were indexed using SAMTools (v1.16.1). Variant calling was performed using GATK (v4.2.6.1) (McKenna et al, 2010) based on GATK Best Practices. Variants were analyzed and confirmed manually using IGV (version 11.0.13).

## CUT&RUN

Genome-wide profiling of H3K9me3 occupancy was performed using CUT&RUN as previously described (Morillo et al, 2023). Parasites were prepared for CUT&RUN with double percoll/sorbitol synchronization as described above. Parasites were collected at 14–17 h post-invasion and cryopreserved in 10% DMSO to −80 °C. Cells were washed 2x in 1x PBS prior to completing the protocol as described. The full analysis pipeline can be found at https://github.com/KafsackLab/PfCUTandRUN.

## CSA cytoadherence assay

Assays for adhesion to CSA were performed as previously described (Beeson et al, 1999). Briefly, plastic petri dishes (VWR) were coated overnight with 0.05% Chondroitin sulfate A (CSA from bovine trachea, Sigma-Aldrich) in PBS at 4 °C and blocked with 1% bovine serum albumin in PBS for 1 h. NF54 parasites were synchronized using 5% sorbitol one day before the cytoadherence assay and the assays were performed using trophozoite-infected erythrocytes at >3% parasitemia, at 5% hematocrit in RPMI with 10% human serum. Parasites were added to the CSA-coated petri dishes and incubated for adhesion for 1 h at 37 °C with agitation every 15 min. Then, the unbound cells were gently washed off with RPMI (5–6 times) and the bound cells were fixed with 2% glutaraldehyde and stained with Giemsa. The number of bound infected erythrocytes per mm$^2$ was counted microscopically.

## Labeling of infected erythrocytes and flow cytometry

PAM1.4 IgG reactivity to native VAR2CSA expressed on the surface of infected erythrocytes was assessed by flow cytometry, as previously described (Lopez-Perez and Olsen, 2022). Briefly, 100 μL of erythrocytes infected with synchronized late-stage NF54 parasites at >3% parasitemia and 5% hematocrit, were incubated with PAM1.4 (10 μg/mL) for 30 min at 4 °C. After 3 washes with PBS, cells were incubated with 16 μM Hoechst 33342 and anti-human IgG (Fc specific) FITC conjugated antibody (1:100 dilution, Sigma-Aldrich) for 30 min at 4 °C. Following 3 washes, cells were analyzed on Aurora flow cytometer with SpectroFlo software (Cytek Biosciences). Flow-cytometry data were processed using FlowJo v10.

## Western blotting

Analysis of the *Pf*SAMS knockdown line by Western blot was performed using whole-cell lysates. Late-stage parasites were lysed

using 0.1% saponin and resuspended in 1x Laemmli loading buffer diluted in 1x PBS supplemented with 1x Roche Complete protease inhibitors cocktail, then boiled for 10 min. Proteins were then separated by 12% SDS–PAGE and transferred to a PVDF membrane (Invitrogen). Membranes blocked with 5% milk were then probed with anti-hsp70 (1:5000, StressMarq SPtdCho-186) and anti-*Pf*SAMS-706 (1:2000) primary antibody solutions, followed by anti-rabbit IgG (1:5000, Agilent Dako P0399) secondary antibody. Immunoblots were incubated with the chemiluminescent substrate SuperSignal West Pico PLUS (Thermo Scientific, Cat. No. 34577) following manufacturer directions. Images were obtained using a Bio-Rad ChemiDoc MP Imaging System.

## Data availability

The RNA-seq, whole-genome sequencing and CUT&RUN datasets are available in the following database: NCBI Sequence Read Archive, accession ID number PRJNA1282778.

The source data of this paper are collected in the following database record: biostudies:S-SCDT-10_1038-S44318-026-00751-x.

## Peer review information

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

coordinated transcriptional switching network mediates antigenic variation of
human malaria parasites. Elife 11:e83840

## Acknowledgements

The authors would like to thank Dr. Björn Kafsack for help with computational
analysis of genome and transcriptome data and Dr. Riward Campelo Morillo for
assistance in completion of the CUT&RUN analysis. Genome and cDNA
sequencing were performed by the Genomics Core Laboratory at Weill Cornell
Medicine. This work was supported by the National Institutes of Health (AI
52390 and AI99327 to KWD). KWD is a Stavros S. Niarchos Scholar and a
recipient of a William Randolf Hearst Endowed Faculty Fellowship. FF received
support from the Swiss NSF (Early Postdoc.Mobility grant P2BEP3_191777). JEV
received support from F31 Predoctoral Fellowship F31AI164897 from the NIH.
The Department of Microbiology and Immunology at Weill Medical College of
Cornell University acknowledges the support of the William Randolph Hearst
Foundation. The funders had no role in the study design, data collection and
analysis, decision to publish, or preparation of the manuscript.

## Author contributions

**Joseph E Visone**: Conceptualization; Data curation; Formal analysis; Funding
acquisition; Investigation; Methodology; Writing—original draft; Writing—
review and editing. **Francesca Florini**: Conceptualization; Data curation; Formal
analysis; Funding acquisition; Validation; Investigation; Methodology; Writing
—original draft; Writing—review and editing. **Evi Hadjimichael**:
Conceptualization; Data curation; Formal analysis; Validation; Investigation;
Methodology; Writing—original draft; Writing—review and editing. **Valay
Patel**: Investigation. **Kirk W Deitsch**: Conceptualization; Resources; Data
curation; Formal analysis; Supervision; Funding acquisition; Validation;
Investigation; Methodology; Writing—original draft; Project administration;
Writing—review and editing.

Source data underlying figure panels in this paper may have individual authorship
assigned. Where available, figure panel/source data authorship is listed in the
following database record: biostudies:S-SCDT-10_1038-S44318-026-00751-x.

## Disclosure and competing interests statement

The authors declare no competing interests.

# Expanded View Figures

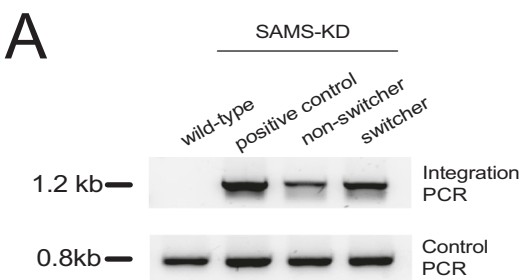

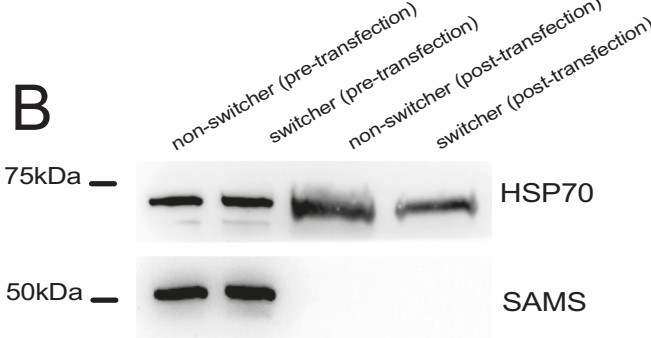

**Figure EV1.   Validation of SAMS-KD parasite lines.**

(**A**) PCR validation confirming correct integration into the target locus. A wild-type, non-transfected line was used as a negative control, and a previously published SAMS-KD line (Harris et al, 2023) served as a positive control. Primer sequences are provided in Dataset EV3. (**B**) Western blot analysis using anti-SAMS and anti-HSP70 antibodies. Protein extracts were prepared from late-stage parasites of a wild-type non-switcher clone and a switcher clone, both before and after transfection with the SAMS-KD construct.

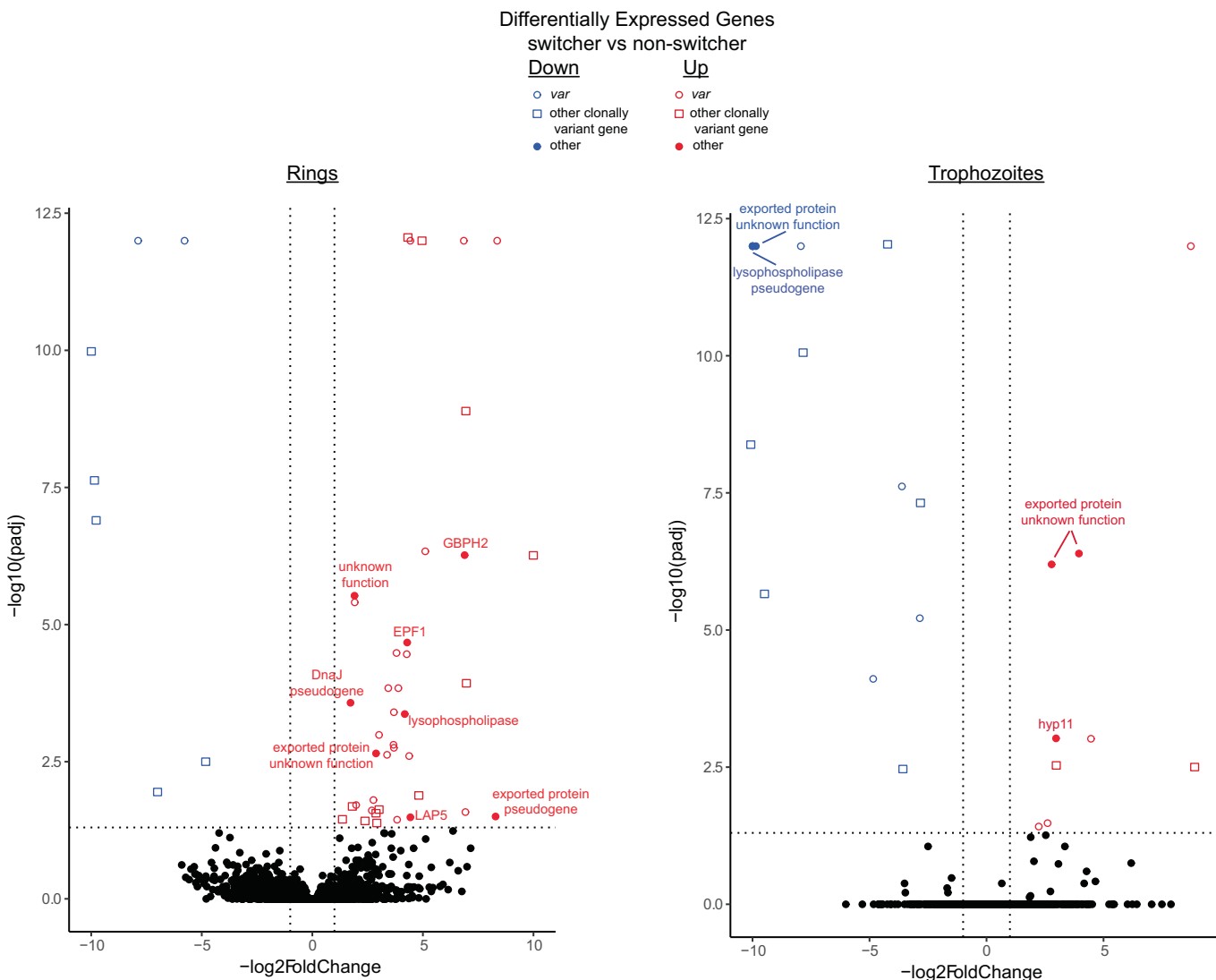

**Figure EV2. Differential gene expression in switcher vs non-switcher lines determined by RNA-Seq.**

Volcano plots displaying differential expression comparing the transcriptomes of switcher and non-switcher parasite populations. Upregulated genes in the switcher line are shown in red while downregulated genes are in blue. *var* genes are displayed as open circles while non-*var* genes from other clonally variant gene families are shown in open boxes. Other genes are labeled with the name of the encoded protein. Full annotation numbers for all genes are included in Dataset EV1. Comparison of ring-stage parasites are shown in the left panel and trophozoites are shown on the right. Differential expressional analysis was performed using DESeq2 (v.1.36) with a false discovery rate cutoff of <0.05. Comparison was performed with two independent replicate transcriptomes for each population. Differentially expressed genes were defined as having a log2foldchange of at least 2.00 and an adjusted p value of less than 0.05.

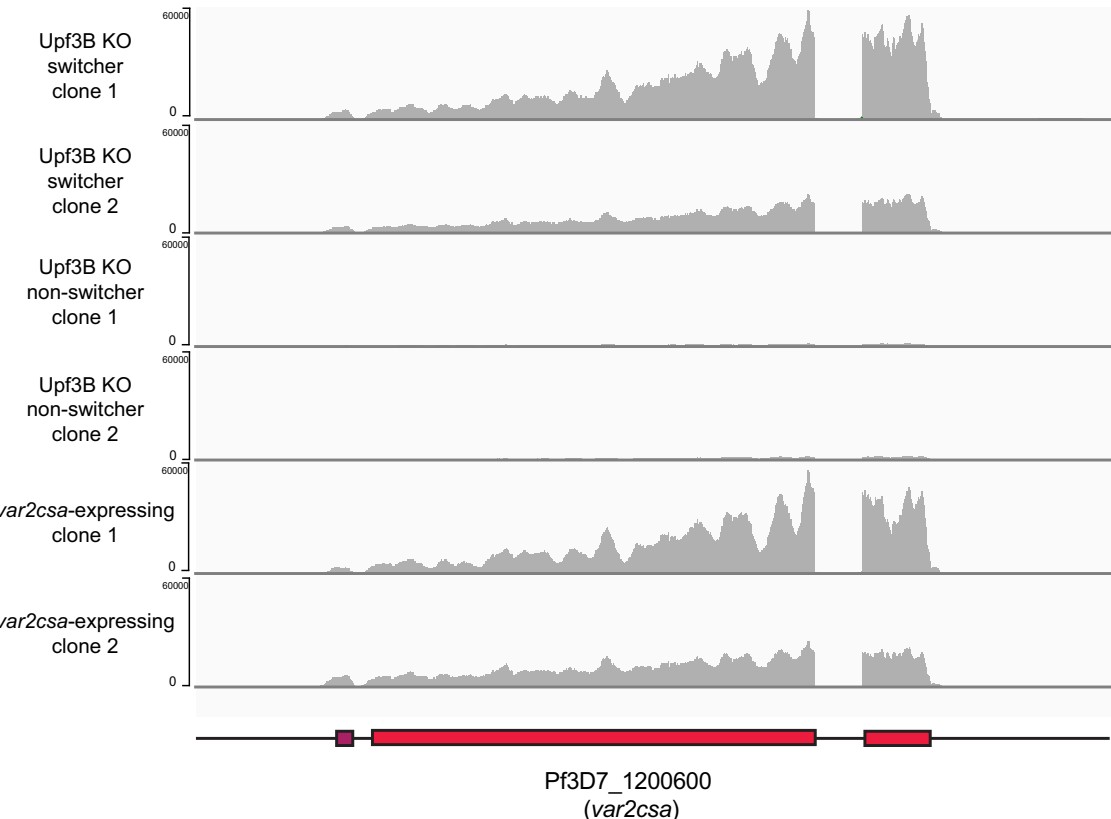

**Figure EV3.  Expression of *var2csa* in wild-type and Upf3B knockout lines detected by RNA-seq.**

RNAseq profiles for the *var2csa* locus in either wild-type lines (bottom 2 profiles) or lines in which the *Pfupf3B* gene has been disrupted in either a switcher (top 2 profiles) or non-switcher (middle two profiles) background. Reads were normalized by downsampling. Each profile represents an independent biological replicate.

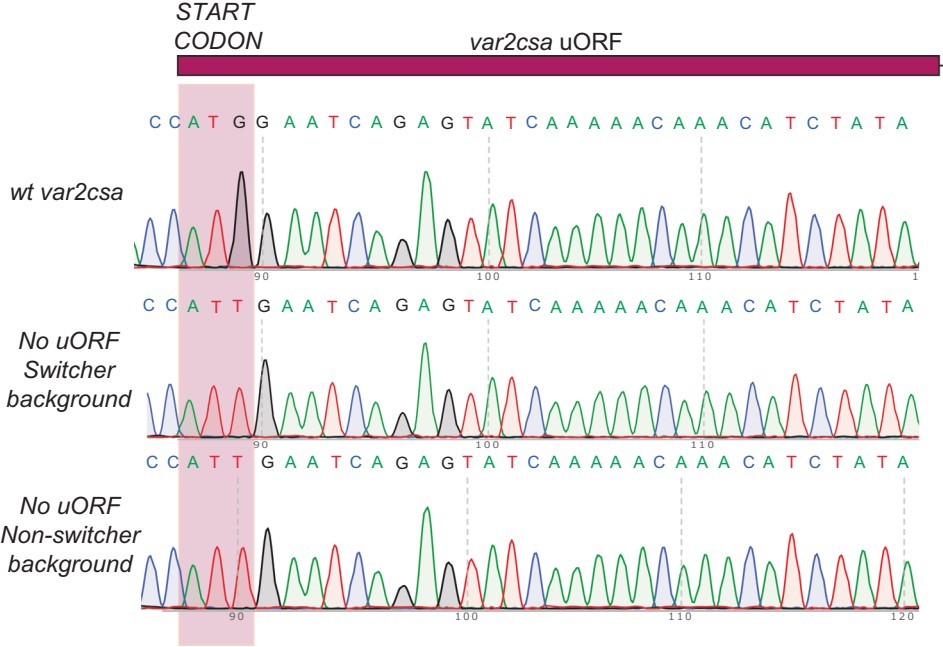

**Figure EV4.  Validation of uORF starting codon modification.**

Sanger sequencing confirming the modification of the *var2csa* uORF start codon from ATG (methionine) to ATT (isoleucine) in both switcher and non-switcher backgrounds. A wild-type, non-transfected line is included as a control. The purple box highlights the three bases encoding the start codon.

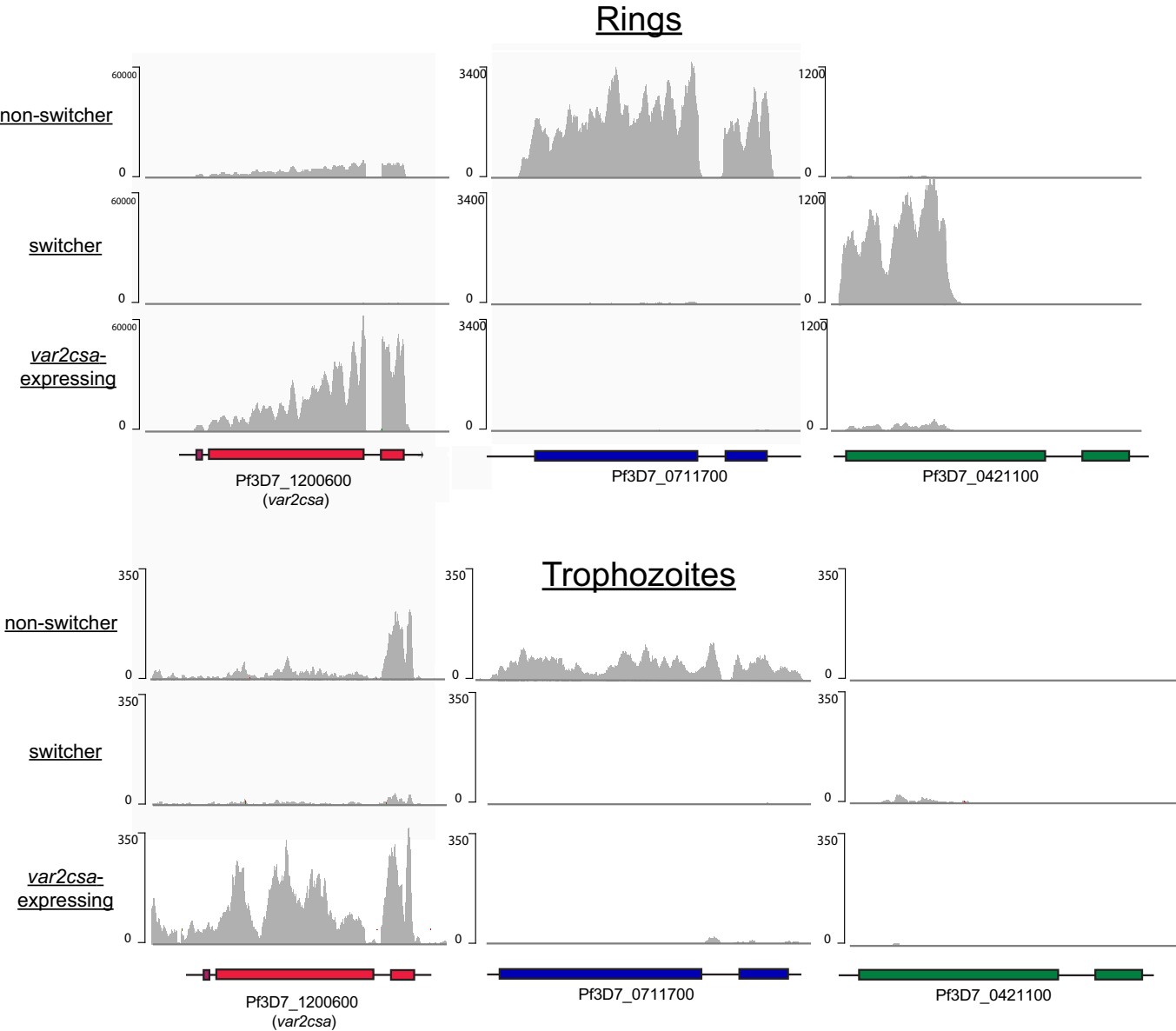

**Figure EV5. Expression of dominant *var* gene in switcher, non-switcher and *var2csa* expressing lines detected by RNA-seq.**

RNAseq profiles for the dominant *var* gene in both rings and trophozoites for a non-switcher (top), switcher (middle) and *var2csa* expressing line. Note that multimapping of reads was not allowed, thus for Pf3D7_0421100, only reads mapping to the 5' end of the gene could be unambiguously mapped and are displayed. Reads mapping to the 3' end of the gene were discarded due to near complete sequence identity elsewhere in the genome, thus preventing unambiguous mapping. Reads were normalized by down-sampling.

