## [Peer Review File · The EMBO Journal]

Mechanistic insights into coordinated var transcriptional switching in malaria parasites

Joseph Visone, Francesca Florini, Evi Hadjimichael, Valay Patel, and Kirk Deitsch

Corresponding author(s): Kirk Deitsch (kwd2001@med.cornell.edu)

Review Timeline:

Submission Date:	2nd Aug 25
Editorial Decision:	28th Sep 25
Revision Received:	12th Dec 25
Editorial Decision:	11th Feb 26
Revision Received:	12th Feb 26
Accepted:	25th Feb 26

Editor: Ioannis Papaioannou

Transaction Report:

Dear Dr. Deitsch,

Thank you again for the submission of your manuscript EMBOJ-2025-122057 for consideration by The EMBO Journal, and for your patience during peer review. Your manuscript has now been seen by three experts in the field, and we have received the full set of their detailed, informative, and constructive reports, which are included below.

I am very pleased to say that the referees recognize that the study addresses a significant gap in our knowledge of the molecular mechanisms driving antigenic variation, and that it provides significant new functional insights into these mechanisms in malaria parasites. The referees find the study well-designed and the conclusions well-supported, for the most part. They also identify a number of gaps and make useful suggestions for further strengthening of the work and increase of its impact on the field.

In light of the largely positive referees' comments and recommendations, I would like to invite you to submit a revised version of your manuscript taking the referees' suggestions on board, along with a detailed point-by-point response addressing all referees' comments. Please note that it is The EMBO Journal policy to allow only a single round of major revision, and acceptance of your manuscript will therefore depend on the completeness of your responses in this revised version.

Please let me know if you have any questions or comments that you would like to discuss with me. You are very welcome to share with me a draft point-by-point response letter/revision plan explaining if there are any points you do not agree with or cannot address, or alternatively we could arrange a video call, if you prefer.

We generally allow three months as standard revision time (December 27, 2025). As a matter of policy, competing manuscripts published during this period will not negatively impact our assessment of the conceptual advance presented by your study. However, we request that you contact us as soon as possible upon publication of any related work, to discuss how to proceed. Should you foresee a problem in meeting this three-month deadline, please let us know in advance and we will be able to grant an extension.

Thank you for the opportunity to consider your work for publication in The EMBO Journal. I look forward to your revision.

Best regards,

Ioannis

Instructions for preparing your revised manuscript

1. When you are ready to submit the revision, please upload:

- A Word file of the manuscript text (including legends of main Figures, EV Figures and Tables). Please make sure that changes are highlighted (or "tracked") to be clearly visible.

- Individual production-quality figure files (one file per figure). When assembling your figures, please refer to our figure preparation guidelines in order to ensure proper formatting and readability in print as well as on screen:

If the data shown in a figure are obtained from n {less than or equal to} 2, please use scatter plots showing the individual data points.

- i. the name of the statistical test used to generate error bars and P values
- ii. the number (n) of independent experiments (please specify technical or biological replicates) underlying each data point (discussion of statistical methodology can be reported in the Materials and Methods section, but figure legends should contain a basic description of n , P, and the test applied)
- iii. the nature of the bars and error bars (s.d., s.e.m.).

- A point-by-point response to the referees' comments, with a detailed description of the changes made (as a word file). All referees' concerns must be fully addressed and their suggestions taken on board. When preparing your letter of response to the referees' comments, please bear in mind that this will form part of the Review Process File and will therefore be available online to the community. Please note that you have the possibility to opt out of the transparent process at any stage prior to publication by letting the editorial office know (contact@embojournal.org); if you do opt out, the Review Process File link will point to the following statement: "No Review Process File is available with this article, as the authors have chosen not to make the review process public in this case.". For more details on our Transparent Editorial Process, please visit our website: <https://www.embopress.org/page/journal/14602075/authorguide#transparentprocess>

- Expanded View (EV) files (replacing Supplementary Information) that are collapsible/expandable online. A maximum of 5 EV Figures can be typeset. EV Figures should be cited as "Figure EV1, Figure EV2" etc. in the text, and their respective legends should be included in the manuscript file after the legends of regular figures. See detailed instructions regarding Expanded View files here: <https://www.embopress.org/page/journal/14602075/authorguide#expandedview>

- For the figures that you do NOT wish to display as Expanded View figures, they should be bundled together with their legends in a single PDF file called "Appendix", which should start with a short Table of Contents (including page numbers). Appendix figures should be referred to in the main text as: "Appendix Figure S1, Appendix Figure S2" etc. Please see detailed instructions here: <https://www.embopress.org/page/journal/14602075/authorguide#expandedview>

- A complete author checklist, which you can download from our author guidelines (<https://www.embopress.org/page/journal/14602075/authorguide>). Please note that the checklist will also be part of the Review Process File.

2. Please note that no statistics should be calculated and shown in Figures if $n=2$. Please also note that each p value should be reported as an exact value.

3. Before submitting your revision, primary datasets (and computer code, where appropriate) produced in this study need to be deposited in appropriate public databases (see <https://www.embopress.org/page/journal/14602075/authorguide#dataavailability>). In particular, all genome and RNA sequencing data generated in your study must be deposited in an appropriate repository. The accession numbers, database, and the specific URLs (links) should be listed in a formal "Data availability" section (placed after Methods), following the example below:

"The RNA-seq datasets produced in this study are available in the following database:
Gene Expression Omnibus GSE46843 (<https://www.ncbi.nlm.nih.gov/geo/query/acc.cgi?acc=GSE46843>)"

*** All links should resolve to a page where the data can be accessed. ***

*** Please remember to provide in the Data availability section of your revised manuscript reviewer passwords if the datasets are not yet public. ***

*** The Data Availability Section is restricted to new primary data that are part of this study. In case you have no data that require deposition in a public database, please state so instead of referring to the database: "Our study includes no data deposited in public repositories." under the heading "Data availability". ***

4. The materials and methods need to be described in the manuscript using our structured methods format, which is now required for all research articles. According to this format, the Methods section includes a single "Reagents and Tools Table" - listing key reagents, experimental models, software and relevant equipment including their sources and relevant identifiers - followed by a "Methods and Protocols" section describing the methods. Please download and fill our Reagents and Tools Table template (.docx), which you can find in our author guide: <https://www.embopress.org/page/journal/14602075/authorguide#structuredmethods>. When submitting your revised manuscript, please do not include the Reagents and Tools Table in the Methods section of the manuscript but instead upload it as a separate file choosing the file type "Reagent Table".

5. Please check that the title and the abstract of the manuscript are brief, yet explicit, even to non-specialists. The length of the title should not exceed 100 characters, and the abstract should be a single paragraph not exceeding 175 words.

6. Please also note our reference format: <https://www.embopress.org/page/journal/14602075/authorguide#referencesformat>.

8. Please remember: digital image enhancement is acceptable practice, as long as it accurately represents the original data and

conforms to community standards. If a figure has been subjected to significant electronic manipulation, this must be noted in the figure legend or in the "Materials and Methods" section. The editors reserve the right to request original versions of figures and the original images that were used to assemble the figure.

9. Our journal encourages inclusion of data citations in the reference list to directly cite datasets that were obtained from public databases. Data citations in the article text are distinct from normal bibliographical citations and should directly link to the database records from which the data can be accessed. In the main text, data citations are formatted as follows: "Data ref: Smith et al, 2001" or "Data ref: NCBI Sequence Read Archive PRJNA342805, 2017". In the Reference list, data citations must be labeled with "[DATASET]". A data reference must provide the database name, accession number/identifiers, and a resolvable link to the landing page from which the data can be accessed at the end of the reference. Further instructions are available at: <https://www.embopress.org/page/journal/14602075/authorguide#referencesformat>.

10. We request authors to consider both actual and perceived competing interests. Please review our policy (<https://www.embopress.org/page/journal/14602075/authorguide#conflictsofinterest>) and update your competing interests statement if necessary. Please name this section 'Disclosure and competing interests statement' and place it after the Acknowledgements section.

11. Please note that all corresponding authors are required to provide an ORCID ID upon submission of a revised manuscript (<https://orcid.org/>). Please find instructions on how to link your ORCID ID to your account in our manuscript tracking system in our Author guidelines (<https://www.embopress.org/page/journal/14602075/authorguide#authorshipguidelines>).

12. We use CRediT to specify the contributions of each author in the journal submission system. CRediT replaces the author contribution section, which should be removed from the manuscript. Please use the free text box to provide more detailed descriptions. See also guide to authors: <https://www.embopress.org/page/journal/14602075/authorguide#authorshipguidelines>.

14. We would also welcome the submission of cover suggestions or motifs to be used by our Graphics Illustrator in designing a cover.

Referee #1:

In this manuscript, the authors addressed a critical molecular mechanism of the deadliest malaria parasite, *Plasmodium falciparum*, that potentially drives the key component of antigenic variation, var gene switching. Specifically, the authors demonstrate that var2csa, a highly conserved member of the otherwise highly variable var gene family, may be the central component of the mechanism that facilitates switching between the expression of individual var gene family members and/or the lack thereof. Here, the authors show that transcription of the var2csa locus varies dramatically between *P. falciparum* lines that exhibit var gene switching and those that do not (the "non-switchers"). This is linked to an lncRNA transcript within the var2csa intron that is highly expressed in the non-switcher lines (NS) but completely suppressed in the switchers (S). The authors suggest that the lncRNA mediates silencing of the var2csa mRNA transcript, which in some way prevents switching between (other) var genes. On the other hand, the derepressed var2csa transcript associated with the lncRNA repression promotes var gene switching. The deactivation of var2csa is, in turn, facilitated by the Nonsense Mediated Decay (NMD), and thus NMD-mediated var2csa degradation prevents switching. This can also be mimicked by suppression of var2csa transcription by deactivation of a uORF upstream of the gene. By several transgenic strategies, the authors showed that stabilization of the var2csa transcript, "one way or another" (Figure 5), consistently blocks switching. In the wild-type parasites, the var2csa transcripts are stabilized by their active translation, which prevents recruitment of the NMD.

Overall, this is a well-conceived set of experiments and well-supported results addressing a highly important knowledge gap about molecular mechanisms driving antigenic variation. As such, I support its publication in the EMBO Journal. Nonetheless, I do have some concerns about some aspects of the presented results regarding their broader implications on one side and the interpretation (or limitation of which) of the obtained data on the other. As I don't believe that more experiments are needed for this manuscript, it would be helpful if the author could comment on these in the text.

Specifically:

1. I am somewhat unclear on the definition of the SWITCHER (S) vs. NON-SWITCHER (NS) phenotype. I do understand that

this was published in earlier studies; however, it would be helpful to provide a better picture of this in this manuscript. Is the difference between the S and NS always so sharp, or are there situations where the switching occurs in an intermediate state? How many genes have to be switched, and how long does it take before a line can be called SWITCHER or not? Moreover, could S and/or NS be derived from many parasite strains and genetic backgrounds with the same frequency? In the future, it would be essential to understand how universal and generally applicable these results are. Such studies are indeed beyond the scope of his study, nevertheless, some comments on this issue would be helpful, either in the Results or Discussion section.

2. In Figure 3, the authors measure the expression of the lncRNA first by RT-PCR in three S and three NS parasite lines. It is unclear what these lines are. Are these isogenic clones to each other? Also, could this method be used to distinguish S and NS lines in other parasite strains? Are these lncRNA levels always so uniform, high in NS, and virtually zero in S? Could there be intermediate states associated with intermediate levels of the phenotypes? In the RNA-Seq, there are some detectable reads over the lncRNA locus. Is there perhaps a threshold between S and NS lines in the lncRNA levels?

3. Investigating the role of MND in the var2csa, the authors generated a deletion line with UPFB3 genes, which is a key NMD factor, deactivating the entire pathway. Although the result demonstrating the stabilization of the var2csa gene is clear, I believe that deactivating the entire NMD will have a far-reaching effect on the parasite's physiology. Could one exclude the possibility that the var2csa transcript stabilization is an indirect effect of NMD deactivation as an overall disturbance of the parasite physiology? Is there a precedent for studying the direct effects of NMD on individual transcript stability by deactivating the whole pathway?

4. I struggle to understand the oval model that the author proposes to mediate the switching vs non-switching state. In the first paragraph of the discussion, the authors suggest that the switching or lack of it is mediated by transcriptional status at the var2csa promoter. However, most of the results presented here do not examine the actual transcriptional status, but rather the steady state of var2csa mRNA, which NMD degrades in conjunction with lncRNA expression in S lines (?). The impression I'm getting is that it's the sheer presence of the var2csa mRNA that mediates switching. It would be good to comment on this somewhat more and perhaps speculate on additional possible mechanisms of var gene silencing in the discussion part.

Referee #2:

- general summary and opinion about the principal significance of the study, its questions and findings

In the current manuscript, Visone et al. set out to decipher the mechanism underlying var gene switching, which resulted in a significant step forward in understanding antigenic variation of *P. falciparum*. They detect that the key difference between clones with differing switching phenotypes is the expression of var2csa. They then elucidate how these different states of var2csa expression and translation are linked to the expression of an intron derived ncRNA and the NMD-pathway that is triggered by an uORF.

The authors have produced a range of genetic backgrounds to test their different hypothesis and it was overall easy to follow their logic and journey of discovery, which also fits with previous work of the same team. Most importantly, their results nicely consolidate previous findings on var gene switching and var2csa biology into one coherent model.

- specific major concerns essential to be addressed to support the conclusions

Overall, the authors provide solid experimental evidence for the role of the upstream open reading frame, the intron ncRNA and the NMD pathway between different var gene switching phenotypes.

I have no major experimental concerns that would have to be addressed to support the conclusions of their data. Please see my comments on some aspects of data interpretation below.

- minor concerns that should be addressed

Are the var2csa transcript levels in NMD-disrupted parasites similar to those in the 'WT' parasites stably expressing and translating var2csa?

Do the authors have any indication on the relative amounts of switcher, non-switcher and var2csa cells in the field?

The authors describe how the particular var2csa biology can facilitate the switching from one var gene to another one. With the presented model, the authors propose a mechanism for this, yet at the same time also transfer the switching phenotype to another level, i.e. how can var2csa switch from one state to another? Can the authors elaborate on this point and/or share their thoughts/hypotheses?

In the model (Figure 7b), the authors describe that in a switcher phenotype, the promoter of var2csa can compete with that of a different var gene. Figure 2c shows that var2csa transcription and that of the active var gene overlap at ~4-7hpi, suggesting that those two loci can be in a non-heterochromatinized state at the same time that allows for competition. How would that work with a new var gene, that should (as far as we know) be fully covered by H3K9me3/HP1?

Similarly, the authors mention in their model that the presence of the placenta defines whether the var2csa is translated and

stabilized or degraded. How do the authors suggest that 'sensing' the presence of the placenta occurs? Wouldn't this require some baseline translation of var2csa even in the switcher cell line in order to export PfEMP1-var2csa as a 'test balloon' and subsequent selection in case the placenta is present?

Please define NMTGS the first time it is used in the manuscript

Seb Baumgarten

Referee #3:

General summary and opinion about the principal significance of the study, its questions and findings

In this study, Vinsone and Florini et al. made use of previously generated clonal lines of *P. falciparum* that differ in their ability to undergo transcriptional switching of PfEMP1, the major pathogenicity factor encoded by the var gene family and central to immune evasion. Their aim was to investigate the role of the interstrain-conserved var2csa gene within the structured switching network.

They demonstrated that in the non-switching parasite line, the var2csa intron is depleted of the heterochromatin marker H3K9me3. This depletion leads to elevated levels of exon 2-derived sense ncRNA, which had previously been linked to var gene silencing. As a consequence, var2csa is not activated, thereby abolishing its function as a switching intermediate and resulting in a non-switching phenotype. In contrast, in the clonal line that does exhibit progressive changes in var gene expression over time, var2csa expression is detectable-potentially due to inactivation of the var2csa intron promoter. The authors further show that degradation of var2csa transcripts via the NMD pathway is essential for transient expression and subsequent switching away from var2csa, thus enabling antigenic variation. Supporting this model, they performed elegant experiments demonstrating that UPF3 knockout (an NMD regulator) or parasite with non-functional uORF (mutated TSS, elongation) get stuck on var2csa expression.

Overall, this study provides exciting functional insights into the molecular mechanisms of antigenic variation in malaria parasites and highlights the unique role of the conserved var2csa gene. However, it remains unresolved how parasite lines with low var2csa activation rates and thus lacking convergence toward var2csa achieve switching.

Specific major concerns essential to be addressed to support the conclusions

1. Most experiments were performed using only a single subclonal switcher and non-switcher line, even though the laboratory had previously published the generation of multiple clonal lines with both phenotypes. At least key experiments such as the expression of var2csa in switching clonal lines and the absence of H3K9me3 in the var intron in non-switchers should be shown for at least three biological replicates. This is important because the variants studied are members of different var gene groups (B/C and C) and also differ in their chromosomal environment, e.g., chromosome, type of upstream sequence, and presence of a ruf6 ncRNA in head-to-head orientation. Can the authors rule out that the phenotypes they describe are inherent characteristics of the different var groups? Could it be, for example, a specific feature of var group C that it exhibits less var2csa activation, since it is known that type C var genes are frequently expressed in *in vitro* culture due to their low off rate?
2. The proposed concept of transient activation of var2csa is still unclear. What is meant by "these transcripts were transient in nature" (line 228ff) - since almost all Pf genes are regulated over the IDC. Does a subpopulation of parasites express var2csa in order to switch to a "regular" var gene (new or old variant) during the next IDC? Or is var2csa expressed earlier (5-7 hpi) in switching-capable parasites and another var gene variant later (7-19 hpi) in the same IDC? scRNAseq should provide information on whether a significant proportion of cells express both genes, which could be the case due to the temporal overlap of both expression profiles. The data may already be contained in the scRNAseq dataset from Florini et al. (2025).

Minor concerns that should be addressed

- Lines 22ff, 53f, 61ff, 79ff: Formally, antigenic variation of *P. falciparum* has never been shown in human infections and is still just a hypothesis. Furthermore, it is currently unknown (i) whether a single variant is actually expressed dominantly in all parasites contributing to a parasitemia peak and (ii) whether successive parasitemia peaks are the result of transcriptional switching and/or immune selection of a diverse parasite population. For example, it has been shown that many different subtelomeric B-type variants are expressed simultaneously by the parasite population in the first parasitemia peak, which also contradicts the statement that only a small number of genes are expressed at any given time.
- Line 64ff: Reference missing for the length of an infection.
- Line 103ff: This is an oversimplification, as var1, for example, is even more conserved than var2csa, but its function is still unclear. And what about var3 variants.
- Line 107: P missing in PfEMP1
- Line 132-136: These observations need to be referenced.
- Line 227: The expression of var2csa at an earlier stage in the IDC has been observed before (as cited in line 222), so please clarify the statement "a phenomenon not previously observed".
- Figure 2: One of the key questions here is not addressed: Do the switcher and non-switcher lines differ in their ability to bind

CSA? Or in the time required to enrich parasites for CSA binding?

- Line 267ff: According to the results section, qPCR and RNAseq data were generated from trophozoite stage and "tightly synchronized late-stage parasites" (line 271ff), respectively. However, the "Materials and Methods" section only mentions that parasites in the ring stage were used for qPCR, and RNAseq was performed with parasites in the ring stage (16-19 hpi) and late stage (32-35 hpi). Please clarify this! And if already available, please also show the RNAseq data from parasites in the ring stage for the dominantly expressed variant and the var2csa locus.

- Figure 3:

(1) Panel A: Parasites aged 14-17 hpi (= ring stage) were used for the CUT&TAG experiment, but the intron promoter produces sense lncRNA from exon 2 in late-stage parasites. This could mean that there is a difference in the as lncRNA. Did the authors also check this? For comparison, the H3K9me3 profiles and quantification of the lncRNAs of the dominantly expressed var genes should also be shown. Please note that the var2csa locus is located on the left arm of chromosome 12 and not on the right where the box is shown.

(2) Panel C: Data from three biological replicates are shown here. Unfortunately, it is not mentioned which var genes are dominantly expressed in these cell lines and how long they were monitored to determine the switching/non-switching phenotype. How does lncRNA expression look in the var2csa-expressing parasites from panel A? The statistical test used here is not appropriate (see comment on the calculation of relative expression).

(3) Panel D: Even if parasites in the trophozoite stage are used, how do you explain that the non-switcher line has more var2csa exon 1 expression than the switcher line, which contradicts Figure 1? What about the var2csa-expressing line? cDNA synthesis was performed using an oligodT and exon 2 primer mix, whereby the var2csa primer is specific for a region in the middle of exon 2. However, this is not apparent in the coverage plot (reads map to the entire exon 2). Is there an explanation for this?

(4) Panel E: The title says "var2csa s-lncRNA (RNA-Seq), which should be corrected to "var2csa exon 2 reads" (or something similar) as the approach is not able to discriminate between exon 2 reads originating from the upstream or the intron promoter. For three replicates a Mann-Whitney-U test (not assuming a normal distribution of the data) should be performed to assess statistical significance.

- Line 286ff: According to the RNAseq data var2csa is not completely silenced in the non-switcher line and Figure 4D also shows that the phenotype is not 100% stable, so the statement needs to be weakened.

- Line 306ff: According to Hugh et al., UPF3B lacks the EJC motif because it is truncated at the C-terminus. Since there is not much data on NMD in plasmodia, it would be interesting to know whether *P. falciparum* has an EJC-independent NMD, such as the ciliate *T. thermophila*. Please comment! Have you also looked at the other key NMD proteins, UPF1 and UPF2?

- Line 323ff: Are the parasites affected in any way by the absence of uORF transcription/translation? Please comment!

- Line 318ff: Please indicate where the whole genome sequencing data for the UPF-KO cell lines can be found!

- Figure 4: How long were the different parasite lines monitored (time between the left and right pie charts in panels C and D)? Did you analyze the parasites in the non-switcher line (var2csa and exon 2 transcription, H3K9me3) that switched expression to var2csa (see panel D)?

- Where is the data showing the statement in line 392ff: "Further, when the var2csa transcript cannot be degraded, parasites appear to be unable to switch away from transcribing the var2csa locus."

- The role of PTEF could be interesting, but has not been further investigated, e.g., through KD, KO, or KS approaches and var gene expression analyses?

- Line 406ff, Figure 6A: The figure does not show a correlation between PTEF and var2csa expression, as stated in the text, only bar charts with pairwise comparisons. Again, a t-test was performed for non-linear relative expression values and no correction was made for multiple testing.

- Figure 6B-D: It is not surprising that WT var2csa-expressing parasites also display VAR2CSA on their surface. For me, it would have been much more informative to analyze switcher and non-switcher lines with differences in transient var2csa activation (see comment above).

- Figure 7: The legend indicates the panels A to C, but numbers are given in the figure. Please explain the abbreviation NMTGS (also only as an abbreviation in the main text, line 444).

- Line 560ff: How were the modified parasites selected? Subcloning?

- Line 577: Typing, should be TRizol, not TRiZol

- Line 579ff: How was the cDNA synthesized? Using oligodT or random hexamers? Delete the space between "Super" and "Script", the correct name is SuperScript II.

- The PCR mixture and cycle conditions for qPCR should be mentioned. Quantification using $2^{\Delta\Delta Ct}$ requires not only equal amplification efficiency for normalizer and gene of interest ($=2$), but also for all var gene primers when comparing var gene X with var gene Y. Has this been verified experimentally?

- Line 604: Please mention the specificity (all var genes) and binding region of the primers used for exon 2 enrichment!

- Line 605ff: Is the indicated Illumina kit, Illumina DNA Prep Kit, correct for RNAseq?

- Line 608ff: The STAR argument basically excludes all reads with more than one mapping position, or in other words, only uniquely mapped reads were allowed. I would state this very clearly, as it increases the confidence that no exon 2 reads from other var (pseudo) genes are shown in Figure 3. Not var genes are multimapped, but their reads.

- Line 617ff: Please comment on the read length, the strain-specificity, the initial and down-sampled read depth.

- Line 584ff: The formula for calculating relative expression assumes equal amplification efficiency for the normalizer and the gene of interest ($=2$) and does not produce values with a linear scale, e.g., $2^2 = 4$ and $2^{-2} = 0.25$ (requires log10 transformation), Therefore, it is not valid to perform a t-test on the data, as shown in several figures.

Any additional non-essential suggestions for improving the study (which will be at the author's/editor's discretion)

- In previous studies, the authors have shown subclonal lines with a so-called "many" phenotype. How do these parasites behave in terms of the expression of var2csa and exon 2 ncRNA, etc.?
- Var2csa exon 2 pseudogenes (clade A-ATS) terminate 5 of 7 internal Var gene clusters on the opposite DNA strand (Otto et al. 2018). Do you think they have a function within the switching network?

Dear Dr. Papaioannou,

Thank you again for handling our manuscript EMBOJ-2025-122057 for consideration by The EMBO Journal. We were very pleased to see that the referees recognize that the study addresses a significant gap in our knowledge of the molecular mechanisms driving antigenic variation, and that it provides significant new functional insights into these mechanisms in malaria parasites. We would like to thank the reviewers for their very thorough and complete review of the paper and for their detailed suggestions. They clearly spent significant time and effort preparing their reviews, which we believe have helped us improve the paper.

Given that the reviewers did not suggest any additional experiments, we have focused our efforts on improving clarity and providing additional analysis. This includes improving the text, slightly modifying some of the figures and correcting some of the statistical analysis. We also added several extended view figures and tables to provide the analysis requested by the reviewers. Overall, we were able to address all the questions, comments and criticisms from the reviewers.

Below we have included a point-by-point response to all the reviewers' comments. We have included the original comment as well as our response (in **blue text**). In addition, we have included in **bold text** the page number or figure/table that includes the changes. These numbers are with reference to the version of the manuscript with **changes tracked**.

We hope the manuscript is now acceptable for publication in the EMBO Journal.

Thanks again for handling the manuscript. Please don't hesitate to contact me if any additional information is required.

Best regards,

Kirk Deitsch

Referee #1:

In this manuscript, the authors addressed a critical molecular mechanism of the deadliest malaria parasite, *Plasmodium falciparum*, that potentially drives the key component of antigenic variation, var gene switching. Specifically, the authors demonstrate that var2csa, a highly conserved member of the otherwise highly variable var gene family, may be the central component of the mechanism that facilitates

switching between the expression of individual var gene family members and/or the lack thereof. Here, the authors show that transcription of the var2csa locus varies dramatically between *P. falciparum* lines that exhibit var gene switching and those that do not (the "non-switchers"). This is linked to an lncRNA transcript within the var2csa intron that is highly expressed in the non-switcher lines (NS) but completely suppressed in the switchers (S). The authors suggest that the lncRNA mediates silencing of the var2csa mRNA transcript, which in some way prevents switching between (other) var genes. On the other hand, the derepressed var2csa transcript associated with the lncRNA repression promotes var gene switching. The deactivation of var2csa is, in turn, facilitated by the Nonsense Mediated Decay (NMD), and thus NMD-mediated var2csa degradation prevents switching. This can also be mimicked by suppression of var2csa transcription by deactivation of a uORF upstream of the gene. By several transgenic strategies, the authors showed that stabilization of the var2csa transcript, "one way or another" (Figure 5), consistently blocks switching. In the wild-type parasites, the var2csa transcripts are stabilized by their active translation, which prevents recruitment of the NMD.

Overall, this is a well-conceived set of experiments and well-supported results addressing a highly important knowledge gap about molecular mechanisms driving antigenic variation. As such, I support its publication in the EMBO Journal. Nonetheless, I do have some concerns about some aspects of the presented results regarding their broader implications on one side and the interpretation (or limitation of which) of the obtained data on the other. As I don't believe that more experiments are needed for this manuscript, it would be helpful if the author could comment on these in the text.

Specifically:

1. I am somewhat unclear on the definition of the SWITCHER (S) vs. NON-SWITCHER (NS) phenotype. I do understand that this was published in earlier studies; however, it would be helpful to provide a better picture of this in this manuscript. Is the difference between the S and NS always so sharp, or are there situations where the switching occurs in an intermediate state? How many genes have to be switched, and how long does it take before a line can be called SWITCHER or not? Moreover, could S and/or NS be derived from many parasite strains and genetic backgrounds with the same frequency? In the future, it would be essential to understand how universal and generally applicable these results are. Such studies are indeed beyond the scope of his study, nevertheless, some comments on this issue would be helpful, either in the Results of Discussion section.

We agree with the reviewer that these are important and interesting questions. This topic is addressed to some extent in the Discussion where we mention that parasites

can transition between the “switcher” and “non-switcher” phenotypes, presumably through changes in the activity of the intron promoter that drives expression of the ncRNA. We have expanded this section to address several of the questions posed by the reviewer (**see lines 558-568**).

2. In Figure 3, the authors measure the expression of the lncRNA first by RT-PCR in three S and three NS parasite lines. It is unclear what these lines are. Are these isogenic clones to each other?

We agree that our original text did not adequately define these lines. We now explicitly state that these data were obtained from three isogenic, recently subcloned parasite lines derived from the “switcher” and “non-switcher” populations (**see line 277-279**).

Also, could this method be used to distinguish S and NS lines in other parasite strains? Are these lncRNA levels always so uniform, high in NS, and virtually zero in S? Could there be intermediate states associated with intermediate levels of the phenotypes? In the RNA-Seq, there are some detectable reads over the lncRNA locus. Is there perhaps a threshold between S and NS lines in the lncRNA levels?

The reviewer raises an interesting point here that we cannot precisely address with the experimental design employed in this manuscript. Our model is that an individual parasite can be in either the “switcher” or “non-switcher” state, and thus either express or not express the *var2csa* ncRNA. However, because parasites can transition between these two states, a population will contain individual parasites in both states, with the proportions depending on how recently the population was subcloned and which state the originally isolated parasite was in. For the experiments shown in Figure 3, we used recently subcloned parasite lines so that they would be relatively homogenous, although as the reviewer noted, low levels of ncRNA can still be detected in the “switcher” population in the RNA-seq data. We have expanded our description of these data to more fully explain our interpretation (**see lines 291-295**).

The reviewer suggests an interesting possibility that there might be a “threshold” of ncRNA expression that distinguishes the “switcher” and “non-switcher” state, which could be consistent with the data shown in Figure 3. This is a provocative idea but would require analysis of data acquired at the single cell level, which we don’t currently have. This would also require modifying current single cell RNA-seq protocols to ensure capture of non-polyadenylated ncRNAs. These are experiments we hope to conduct in the future.

3. Investigating the role of MND in the *var2csa*, the authors generated a deletion line with UPFB3 genes, which is a key NMD factor, deactivating the entire pathway.

Although the result demonstrating the stabilization of the *var2csa* gene is clear, I believe that deactivating the entire NMD will have a far-reaching effect on the parasite's physiology. Could one exclude the possibility that the *var2csa* transcript stabilization is an indirect effect of NMD deactivation as an overall disturbance of the parasite physiology? Is there a precedent for studying the direct effects of NMD on individual transcript stability by deactivating the whole pathway?

Similar to the reviewer, we anticipated that inactivation of the NMD pathway would have broad ranging effects on the physiology of the parasite. We were therefore surprised that we could detect no change in parasite growth rates or any meaningful changes to the overall transcriptome (with the obvious exception of *var2csa* expression), indicating that the loss of NMD has little to no discernable effect on the parasite (see **Table EV4**). A recent publication from McHugh et al (*mSphere*, 2023) reached a similar conclusion. We agree that the effect on *var2csa* mRNA stability might be indirect, however the experiments employing a uORF start codon mutation are consistent with a direct effect. It is possible that the NMD pathway plays an important or essential role in other stages of the parasite's lifecycle when regulation mRNA stability is prominent, for example during the gamete to ookinete transition. We have added a paragraph to the Discussion to address these ideas (see lines 495-508).

4. I struggle to understand the oval model that the author proposes to mediate the switching vs non-switching state. In the first paragraph of the discussion, the authors suggest that the switching or lack of it is mediated by transcriptional status at the *var2csa* promoter. However, most of the results presented here do not examine the actual transcriptional status, but rather the steady state of *var2csa* mRNA, which NMD degrades in conjunction with lncRNA expression S lines (?). The impression I'm getting is that it's the sheer presence of the *var2csa* mRNA that mediates switching. It would be good to comment on this somewhat more and perhaps speculate on additional possible mechanisms of var gene silencing in the discussion part.

We agree with the reviewer that the model we propose in Figure 7 is somewhat narrowly focused on how *var* mRNA promoters are regulated and does not incorporate many of the additional attributes mentioned by the reviewer. While we would enjoy offering extensive additional speculation about possible mechanisms, we are inclined to maintain some degree of conciseness to the Discussion. Therefore, we have extended the Discussion to briefly mention the ideas offered by the reviewer, including additional citations, and thus to reinforce the idea that additional aspects of *var* gene regulation undoubtedly apply (see lines 489-494).

Referee #2:

- general summary and opinion about the principal significance of the study, its questions and findings

In the current manuscript, Visone et al. set out to decipher the mechanism underlying var gene switching, which resulted in a significant step forward in understanding antigenic variation of *P. falciparum*. They detect that the key difference between clones with differing switching phenotypes is the expression of *var2csa*. They then elucidate how these different states of *var2csa* expression and translation are linked to the expression of an intron derived ncRNA and the NMD-pathway that is triggered by an uORF.

The authors have produced a range of genetic backgrounds to test their different hypothesis and it was overall easy to follow their logic and journey of discovery, which also fits with previous work of the same team. Most importantly, their results nicely consolidate previous findings on var gene switching and *var2csa* biology into one coherent model.

- specific major concerns essential to be addressed to support the conclusions

Overall, the authors provide solid experimental evidence for the role of the upstream open reading frame, the intron ncRNA and the NMD pathway between different var gene switching phenotypes.

I have no major experimental concerns that would have to be addressed to support the conclusions of their data. Please see my comments on some aspects of data interpretation below.

- minor concerns that should be addressed

Are the *var2csa* transcript levels in NMD-disrupted parasites similar to those in the 'WT' parasites stably expressing and translating *var2csa*?

Examination of the RNAseq datasets indicate that the level of *var2csa* transcripts in the NMD knockout lines is very similar to what is observed in wildtype parasites that are stably expressing *var2csa*. We now mention this in the text (**see lines 340-341**) and provide a figure (**Figure EV3**).

Do the authors have any indication on the relative amounts of switcher, non-switcher and *var2csa* cells in the field?

The reviewer asks a very interesting question that we would similarly love to address. Given what we currently know about the three states, to definitively determine which

state individual parasites are in will require single cell data from late-stage parasites to look for the presence or absence of the *var2csa*-derived ncRNA. As far as we are aware, this type of dataset is not currently available from any field samples. However, several datasets from clinical samples have reported that detection of *var2csa* transcripts in non-pregnant individuals is extremely common. Given the data we report here, these transcripts are likely indicative of the early peak of *var2csa* mRNA displayed by “switchers” (Figure 2) and thus indicate the presence of “switchers” in most, if not all, infections. We now mention these observations on **lines 542-547** of the Discussion section.

The authors describe how the particular *var2csa* biology can facilitate the switching from one *var* gene to another one. With the presented model, the authors propose a mechanism for this, yet at the same time also transfer the switching phenotype to another level, i.e. how can *var2csa* switch from one state to another? Can the authors elaborate on this point and/or share their thoughts/hypotheses?

The first comment from Reviewer 1 similarly mentioned this topic and we agree that these are extremely interesting and important questions. We've added additional hypotheses on these topics to the last paragraph of the Discussion section (**see lines 565-575**).

In the model (Figure 7b), the authors describe that in a switcher phenotype, the promoter of *var2csa* can compete with that of a different *var* gene. Figure 2c shows that *var2csa* transcription and that of the active *var* gene overlap at ~4-7hpi, suggesting that those two loci can be in a non-heterochromatinized state at the same time that allows for competition. How would that work with a new *var* gene, that should (as far as we know) be fully covered by H3K9me3/HP1?

The reviewer describes a very interesting question about the precise molecular nature of *var* gene choice that we cannot fully answer with the methods we currently have available to us. As noted by the reviewer, at the initiation of a switching event, the model proposes that promoter competition occurs between the previously active *var* gene and *var2csa*, and our current model of epigenetic regulation suggests that these two genes would be in non-heterochromatin. If the same model holds for switching to a new *var* gene, presumably that gene would similarly need to be in non-heterochromatin, thus enabling competition with the *var2csa* promoter (e.g. three promoters in non-heterochromatin). To validate this model, we would need to be able to determine the chromatin state of all *var* promoters at the single cell level, which currently cannot be done. However, our previously published single cell RNAseq data are consistent with this model. For example, we were frequently able to detect individual cells with transcripts from two *var* genes, indicating that the promoters from both *var* genes were

in open chromatin. If these parasites were undergoing switching (and thus were “switchers”), we infer that the promoter of *var2csa* was also in open chromatin, although we would not detect transcripts due to their degradation by NMD. It is also possible (or likely) that the opening of the chromatin at each promoter occurs sequentially rather than simultaneously during the switching process. We now briefly mention this concept in the Discussion (**see lines 475-480**). What opens the heterochromatin at each promoter is a key question that we hope to be able to address in future work.

Similarly, the authors mention in their model that the presence of the placenta defines whether the *var2csa* is translated and stabilized or degraded. How do the authors suggest that 'sensing' the presence of the placenta occurs? Wouldn't this require some baseline translation of *var2csa* even in the switcher cell line in order to export PfEMP1-*var2csa* as a 'test balloon' and subsequent selection in case the placenta is present?

How parasites sense a placenta and begin to translate the PfEMP1 encoded by the *var2csa* transcript is a fascinating question we are actively pursuing. One possibility that the reviewer suggests is that parasites translate a low level of the protein to enable selection when a placenta is available (the “test balloon”). While we cannot rule this out, we currently favor an alternative model in which the environment of the placenta alters the efficiency of translation reinitiation, thus enabling expression of the VAR2CSA protein. This model is consistent with previous work from our lab showing that translational reinitiation is required for VAR2CSA expression as well as similar examples from model organisms in which the repressive effects of uORFs are overcome by changes in the environment. In these examples, a simple shift in modifications to translation initiation factors can relieve translational repression, often through a stress response pathway or something similar. We have added a short mention of this hypothesis to the Discussion (**see lines 483-486**).

Please define NMTGS the first time it is used in the manuscript

We added this definition and also added citations to this pathway (**see lines 318 and 471**).

Seb Baumgarten

Referee #3:

General summary and opinion about the principal significance of the study, its questions and findings

In this study, Vinsone and Florini et al. made use of previously generated clonal lines of *P. falciparum* that differ in their ability to undergo transcriptional switching of PfEMP1, the major pathogenicity factor encoded by the var gene family and central to immune evasion. Their aim was to investigate the role of the interstrain-conserved var2csa gene within the structured switching network.

They demonstrated that in the non-switching parasite line, the var2csa intron is depleted of the heterochromatin marker H3K9me3. This depletion leads to elevated levels of exon 2-derived sense ncRNA, which had previously been linked to var gene silencing. As a consequence, var2csa is not activated, thereby abolishing its function as a switching intermediate and resulting in a non-switching phenotype. In contrast, in the clonal line that does exhibit progressive changes in var gene expression over time, var2csa expression is detectable-potentially due to inactivation of the var2csa intron promoter. The authors further show that degradation of var2csa transcripts via the NMD pathway is essential for transient expression and subsequent switching away from var2csa, thus enabling antigenic variation. Supporting this model, they performed elegant experiments demonstrating that UPF3 knockout (an NMD regulator) or parasite with non-functional uORF (mutated TSS, elongation) get stuck on var2csa expression. Overall, this study provides exciting functional insights into the molecular mechanisms of antigenic variation in malaria parasites and highlights the unique role of the conserved var2csa gene. However, it remains unresolved how parasite lines with low var2csa activation rates and thus lacking convergence toward var2csa achieve switching.

Specific major concerns essential to be addressed to support the conclusions

1. Most experiments were performed using only a single subclonal switcher and non-switcher line, even though the laboratory had previously published the generation of multiple clonal lines with both phenotypes. At least key experiments such as the expression of var2csa in switching clonal lines and the absence of H3K9me3 in the var intron in non-switchers should be shown for at least three biological replicates. This is important because the variants studied are members of different var gene groups (B/C and C) and also differ in their chromosomal environment, e.g., chromosome, type of upstream sequence, and presence of a ruf6 ncRNA in head-to-head orientation. Can the authors rule out that the phenotypes they describe are inherent characteristics of the different var groups? Could it be, for example, a specific feature of var group C that it exhibits less var2csa activation, since it is known that type C var genes are frequently expressed in in vitro culture due to their low off rate?

We apologize for a lack of clarity on this point in the original version of the manuscript. The Q-PCR data shown in Figure 2 and Figure 3C were in fact generated from three

independent, biological replicates. For example, the early peak of *var2csa* was observed in three independent experiments using different clonal lines. We have changed the text of the figure legends to make this clear. In addition, the expression of *var2csa* exon2 transcripts detected by Q-PCR in three independent lines (Figure 3C) was validated by duplicate RNAseq datasets generated independently. Thus, all the data in these figures were validated by multiple independent replicates and using independent methods.

The second question the reviewer asks regarding a potential influence of the promoter type of the dominant *var* gene on whether a population displays the “switcher” or “non-switcher” phenotype is more difficult to answer. While we performed the analyses described in the manuscript using multiple, independently derived clonal lines, we did not specifically choose additional lines that would represent all different *var* promoter types/chromosomal locations, each displaying both the “switcher” and “non-switcher” phenotypes. We agree that this would be an interesting and exceedingly thorough approach, albeit exceptionally labor and resource intensive. However, we do have data that address this question, in part. The experiments shown in Figure 4D show that “non-switchers” can convert to the “switcher” phenotype at a measurable rate, suggesting the phenotype is not strictly determined by the promoter type of the active gene (see Discussion section, **lines 559-565**). We also have preliminary data showing that alterations to the culture environment can similarly convert “non-switchers” to “switchers”, which we are actively investigating in ongoing experiments. Nonetheless, we agree with the reviewer that we cannot rule out a potential role for *var* promoter type in these interactions, a possibility that is particularly relevant if promoter competition plays an important role in switching, as we hypothesize. Therefore, to address the reviewer’s comments, we now describe these possibilities in the Discussion section (**see lines 565-569**).

2. The proposed concept of transient activation of *var2csa* is still unclear. What is meant by “these transcripts were transient in nature” (line 228ff) - since almost all Pf genes are regulated over the IDC. Does a subpopulation of parasites express *var2csa* in order to switch to a “regular” *var* gene (new or old variant) during the next IDC? Or is *var2csa* expressed earlier (5-7 hpi) in switching-capable parasites and another *var* gene variant later (7-19 hpi) in the same IDC? scRNAseq should provide information on whether a significant proportion of cells express both genes, which could be the case due to the temporal overlap of both expression profiles. The data may already be contained in the scRNAseq dataset from Florini et al. (2025).

In response to the reviewer’s comment, we have removed the phrase “transient in nature” to alleviate any confusion this caused. We think it should now be clear that these transcripts are detected early in the cycle but are not detected later, as we now

explicitly state (**see lines 235-237**). To address the reviewer's question about how we envision this early peak of *var2csa* transcription contributing to switching, we now provide a more detailed description of our model (which also addresses comments from reviewer 2) (see Discussion section, **lines 475-480**). Specifically, we imagine *var* promoters competing for activation and producing transcripts in the same IDC, which we in fact do detect in our previously published scRNAseq dataset (Florini et al, 2025). However, this dataset was generated from tightly synchronized parasites 18-20 hrs post invasion, long after the *var2csa* transcripts are gone, preventing us from using this dataset to directly answer the specific question posed by the reviewer. Nonetheless, analysis of this dataset as suggested by the reviewer provides additional evidence for a general promoter competition model, as we now describe in more detail in the revised Discussion section.

Minor concerns that should be addressed

- Lines 22ff, 53f, 61ff, 79ff: Formally, antigenic variation of *P. falciparum* has never been shown in human infections and is still just a hypothesis. Furthermore, it is currently unknown (i) whether a single variant is actually expressed dominantly in all parasites contributing to a parasitemia peak and (ii) whether successive parasitemia peaks are the result of transcriptional switching and/or immune selection of a diverse parasite population. For example, it has been shown that many different subtelomeric B-type variants are expressed simultaneously by the parasite population in the first parasitemia peak, which also contradicts the statement that only a small number of genes are expressed at any given time.

The reviewer makes an interesting point here that derives from the fact that humans are not an experimental system in which the definitive experiments can be easily done. Nonetheless, observation from human infections, historic experiments performed on humans (both prisoner studies and syphilis treatments) as well as rodent and primate experimental models are all consistent with the concept of antigenic variation by *var* gene switching as inferred in the manuscript. This is the consensus in the field. However, we agree with the reviewer that some nuance in these sentences is warranted. We have therefore rewritten the sentences to not overstate the evidence underlying the model. See modifications to **lines 25-26, 54, 64, 84**.

- Line 64ff: Reference missing for the length of an infection.

We have added references, as requested (see sentence beginning on **lines 68-69**).

- Line 103ff: This is an oversimplification, as *var1*, for example, is even more conserved than *var2csa*, but its function is still unclear. And what about *var3* variants.

We have altered the paragraph to no longer give the false impression that *var2csa* is the only conserved *var* gene, and we have included references that describe both *var1* and the type 3 *vars* (**see line 101**).

- Line 107: P missing in PfEMP1

In this sentence, we are referring to the orthologous protein in other species of the *Laverania* clade. Thus, the “Pf” does not apply. The simpler “EMP1” has been used frequently in the literature when referring to this protein in non-falciparum species and is therefore appropriate here.

- Line 132-136: These observations need to be referenced.

We agree with the reviewer and have added several citations throughout this section (**see lines 137-140**).

- Line 227: The expression of *var2csa* at an earlier stage in the IDC has been observed before (as cited in line 222), so please clarify the statement "a phenomenon not previously observed".

Our intention was to highlight that the early expression of *var2csa* is followed by its rapid decline before the standard *var* gene expression peak. We have modified the sentence to make this clear (**see line 235-237**).

- Figure 2: One of the key questions here is not addressed: Do the switcher and non-switcher lines differ in their ability to bind CSA? Or in the time required to enrich parasites for CSA binding?

While the manuscript is focused on transcriptional and epigenetic regulation of *var* gene switching rather than cytoadherence, we agree with the reviewer that the ability to bind to CSA is an interesting question. For many of the 3D7-derived clonal and transgenic lines used in this study, PfEMP1 export to the RBC surface is either inefficient or disrupted (as described in our previous paper, Florini et al, 2025), thus we cannot address this question. However, we also employed some NF54 lines that can display VAR2CSA on the RBC surface (Figure 6). In these experiments, we tested three wildtype, non-*var2csa* expressing NF54 clones for their ability to bind to CSA (Figure 6B). Importantly, clone 1 is a switcher line and displays the early peak of *var2csa*

expression, as shown in **Table EV5** (a non-switcher is also shown for comparison). Regardless of switcher or non-switcher status, we do not detect any binding to CSA in lines that do not stably express *var2csa*, indicating that the early peak of *var2csa* mRNA does not result in the expression of sufficient surface protein to mediate detectable binding (see **lines 438-441**). Similarly, we only detect surface expression of VAR2CSA when the transcript is stably expressed (Figure 6D), suggesting that the brief expression of the transcript early in the cycle does not lead to detectable amounts of protein on the RBC surface.

We did not determine the time required to enrich for CSA binding in our switcher and non-switcher lines. We presume the reviewer is curious if the early peak of *var2csa* transcripts results in any VAR2CSA protein on the RBC surface that might contribute to CSA binding, thereby reducing the time required for enrichment. While we agree that this is a clever idea, we don't think our current experimental tools could resolve this question. Our current work has demonstrated that the "switcher" lines are able to stably activate *var2csa* expression much more rapidly than the "non-switchers", which will translate into more rapid selection for CSA binding regardless of whether the early peak of expression results in VAR2CSA surface expression. Thus, a simple selection experiment would not resolve the question of translation of the early *var2csa* transcripts. We believe the experiments described in the paragraph above more directly address this question.

- Line 267ff: According to the results section, qPCR and RNAseq data were generated from trophozoite stage and "tightly synchronized late-stage parasites" (line 271ff), respectively. However, the "Materials and Methods" section only mentions that parasites in the ring stage were used for qPCR, and RNAseq was performed with parasites in the ring stage (16-19 hpi) and late stage (32-35 hpi). Please clarify this! And if already available, please also show the RNAseq data from parasites in the ring stage for the dominantly expressed variant and the *var2csa* locus.

We agree with the reviewer that the previous Materials and Methods section was incomplete. We have modified it as requested.

We have added the RNAseq plots for the dominantly expressed genes and *var2csa* at both ring and trophozoite stages to the expanded view section, as requested (**Figure EV5**). However, please note that we exclude multi-mapping reads from these plots to ensure that we are accurately assessing individual *var* gene expression. This means that genes with significant regions of sequence identity elsewhere in the genome have many reads excluded. This is particularly evident for the dominantly expressed gene in the "switcher" line, in which only transcripts from the 5' end of the gene can be uniquely mapped. This was documented in our previous publication (Florini et al, 2025).

- Figure 3:

(1) Panel A: Parasites aged 14-17 hpi (= ring stage) were used for the CUT&TAG experiment, but the intron promoter produces sense lncRNA from exon 2 in late-stage parasites. This could mean that there is a difference in the as lncRNA. Did the authors also check this?

The reviewer is asking an interesting question regarding the presence of an antisense ncRNA that has been reported to be expression from the intron promoter of the active *var* gene during ring stages. We gather that the reviewer is curious if expression of this antisense ncRNA is different at the *var2csa* locus in the three lines described in Figure 3. The RNAseq plots for the ring and trophozoite stages are now provided in the extended view section (**Figure EV5**), however we did not perform strand-specific RNAseq, therefore it is not possible for us to distinguish antisense from sense transcripts. There is indeed greater sequence depth near the intron, which could be due to the presence of antisense ncRNAs, however without definitive results, we prefer not to over-interpret these data.

For comparison, the H3K9me3 profiles and quantification of the lncRNAs of the dominantly expressed *var* genes should also be shown.

As mentioned above, we now provide RNAseq profiles for the dominantly expressed genes in all three lines (**Figure EV5**). These data clearly show mRNA expression from the active gene as expected (and consistent with the Q-PCR data), however as described above, the data are not complete due to exclusion of multi-mapped reads. This also prevents precise quantification of the number of reads for any specific locus (including the ncRNAs), although the patterns are clear.

All H3K9me3 datasets are available through the NCBI database and can be analyzed for the entire genome for all clonal lines described in the manuscript. Interestingly, while we observed the expected pattern of H3K9me3 deposition for the *var2csa* locus consistent with previously published studies, we did not observe striking differences for the other active *var* loci. This is also true for the *var2csa* locus in the “switcher” line in which the promoter is active early in the replicative cycle but nonetheless appears to be occupied by H3K9me3 (Figure 3A). Conversations with other laboratories indicate that this has also been observed when researchers have examined H3K9me3 at transcriptionally active *var* promoters other than *var2csa*, suggesting that the simple paradigm that this histone mark alone results in gene silencing is likely incomplete. This is something we are presently investigating in greater detail, however, we currently do not have sufficient data to definitively resolve the issue. Given that the current manuscript is focused specifically on the *var2csa* locus, we prefer to not distract from the conclusions of the paper.

Please note that the *var2csa* locus is located on the left arm of chromosome 12 and not on the right where the box is shown.

We are aware that the conventional orientation of chromosome 12 is inverted from what is shown in Figure 12A. However, we are also aware that this orientation was chosen arbitrarily and is not of biological significance. More importantly, when presenting these data at numerous scientific meetings, we have found that by convention, most readers are accustomed to viewing gene models and transcription from left to right, and often became confused when we displayed the gene in the orientation suggested by the reviewer. We originally intended to display the chromosome in Figure 3A in the conventional orientation as suggested by the reviewer, however to avoid additional confusion and to maintain consistency, we would then have had to similarly reverse the gene orientation in Figures 3B, 3D, 4A, 4B, 4D, 5A, 5B and 7A-C. Given our previous experience in presenting these data, we prefer to instead simply display the chromosome as currently shown in Figure 3A.

(2) Panel C: Data from three biological replicates are shown here. Unfortunately, it is not mentioned which *var* genes are dominantly expressed in these cell lines and how long they were monitored to determine the switching/non-switching phenotype. How does lncRNA expression look in the *var2csa*-expressing parasites from panel A? The statistical test used here is not appropriate (see comment on the calculation of relative expression).

At the reviewer's request, we now state that these are three different, independent clonal lines, each derived from the original "switcher" and "non-switcher" lines shown in Figure 1A. They are expressing the same dominant *var* genes as the parent and the RNA was obtained shortly after subcloning (**see lines 277-280**). The RNAseq data plots for the *var2csa* locus in trophozoites are now included in **Figure EV5**. As can be seen from the plots, full length transcripts representing mRNA are still present at this time point, making it impossible to differentiate mRNA from ncRNA and thereby quantitate the ncRNA expression levels. However, there appears to be no significant increase in exon 2 transcripts when compared to the rest of the gene, suggesting the ncRNA expression is very low in this population, as expected.

We agree with the reviewer regarding the statistical test for these data and have revised how the statistical analysis was performed. The correct analysis is now described in the figure legend.

(3) Panel D: Even if parasites in the trophozoite stage are used, how do you explain that the non-switcher line has more *var2csa* exon 1 expression than the switcher line, which contradicts Figure 1? What about the *var2csa*-expressing line?

Similar to the reviewer, we were initially puzzled by the small increase in the number of transcripts we detect from exon 1 of the *var2csa* locus in the non-switcher line compared to the switcher. We don't have a definitive explanation for this, however it is worth noting that the locus in the switcher line resides in a region with more extensive heterochromatin marked by H3K9me3 (as shown in panel A of this figure), in particular with respect to the intron, which is transcriptionally active in the non-switcher line. Thus, there could be more "leaky" transcription across the locus or alternatively more transcripts derived from the intron promoter in the non-switcher line, resulting in detection of greater amounts of RNA by RNAseq.

At the reviewer's request, we've added a *var2csa*-expressing line to extended view **Figure EV5**, which shows an even greater amount of exon 1 transcripts, which we presume are from mRNAs that continue to be expressed after the peak of expression in late rings. We similarly detect residual mRNAs for the active *var* genes in the switcher and non-switcher lines, as shown the **Figure EV5**.

cDNA synthesis was performed using an oligodT and exon 2 primer mix, whereby the *var2csa* primer is specific for a region in the middle of exon 2. However, this is not apparent in the coverage plot (reads map to the entire exon 2). Is there an explanation for this?

We and others have observed that oligodT will prime reverse transcription even on non-polyadenylated RNAs in *P. falciparum*, presumably due to the high AT-content of the genome, particularly in noncoding regions. For this reason, the ncRNAs from *var* genes are quite easily detected by RNAseq even when only oligodT is used for priming. We added the custom primers to ensure that these transcripts were efficiently detected, however we are not surprised that our sequencing depth includes all of exon 2.

(4) Panel E: The title says "*var2csa* s-lncRNA (RNA-Seq), which should be corrected to "*var2csa* exon 2 reads" (or something similar) as the approach is not able to discriminate between exon 2 reads originating from the upstream or the intron promoter. For three replicates a Mann-Whitney-U test (not assuming a normal distribution of the data) should be performed to assess statistical significance.

We have corrected the figure and the statistical analysis as requested by the reviewer.

- Line 286ff: According to the RNAseq data *var2csa* is not completely silenced in the non-switcher line and Figure 4D also shows that the phenotype is not 100% stable, so the statement needs to be weakened.

We have adjusted the sentence as requested (**see line 306**).

- Line 306ff: According to Hugh et al., UPF3B lacks the EJC motif because it is truncated at the C-terminus. Since there is not much data on NMD in plasmodia, it would be interesting to know whether *P. falciparum* has an EJC-independent NMD, such as the ciliate *T. thermophila*. Please comment! Have you also looked at the other key NMD proteins, UPF1 and UPF2?

The question of whether *Plasmodium* has an EJC-independent NMD is an interesting one, although given that we are not NMD experts, we are not comfortable making any conclusions on this topic. It is also somewhat peripheral to the focus of the current paper. Regarding UPF1 and UPF2, McHugh and colleagues knocked these out in a recent publication and reported a similar phenotype as us, although they did not investigate *var* gene expression. We now cite this paper in the Discussion (**see lines 502-504**).

- Line 323ff: Are the parasites affected in any way by the absence of uORF transcription/translation? Please comment!

We added to the text that loss of the uORF did not have any discernable effect on the parasites, as we expected given that the *var2csa* locus is often silent (**see lines 355-357**).

- Line 318ff: Please indicate where the whole genome sequencing data for the UPF-KO cell lines can be found!

All genome and transcriptome sequence data, including for the UPF3B knockout lines, are available at the NCBI Sequence Read Archive, as noted in the "Data Availability" section (**see lines 762-765**).

- Figure 4: How long were the different parasite lines monitored (time between the left and right pie charts in panels C and D)? Did you analyze the parasites in the non-switcher line (*var2csa* and exon 2 transcription, H3K9me3) that switched expression to *var2csa* (see panel D)?

We have modified the main text (**see lines 338-339 and 356**) and the figure legend to make clear that this analysis was performed as soon as the transgenic parasites were obtained. No time is need for switching to happen since the transcript is immediately stabilized upon elimination of degradation by NMD, as expected. We did not isolate and examine the small proportion of parasites in the non-switcher line that had activated *var2csa* expression. This would require significant effort, and since we know parasites

can switch between states, we anticipate that these represent parasites that have made this transition, which we now comment on (see Discussion **lines 558-565**).

- Where is the data showing the statement in line 392ff: "Further, when the *var2csa* transcript cannot be degraded, parasites appear to be unable to switch away from transcribing the *var2csa* locus."

This sentence refers to the data in Figure 5F, where even in the absence of selection, the population has fully converged to *var2csa* expression immediately upon loss of NMD. To make this inference clear, we now explicitly state that we are referring to these data (see **lines 414-415**).

- The role of PTEF could be interesting, but has not been further investigated, e.g., through KD, KO, or KS approaches and *var* gene expression analyses?

While we agree with the reviewer that the role of PTEF could be interesting, we did not perform any additional experiments on this protein. However, the Wahlgren lab has previously published some of the experiments suggested by the reviewer, including analysis of knockout lines (Chan et al, 2017). We cite this study in the text (see **lines 426-428**) so readers can assess this previous study in the context of our experiments. It might be useful to further investigate a role for PTEF in some of the lines we describe here, which would be suitable for future work.

- Line 406ff, Figure 6A: The figure does not show a correlation between PTEF and *var2csa* expression, as stated in the text, only bar charts with pairwise comparisons. Again, a t-test was performed for non-linear relative expression values and no correction was made for multiple testing.

We agree with the reviewer that we used incorrect terminology in this sentence. We modified the text to simply say that we observed PTEF expression in populations that express *var2csa* (see **lines 431-433**). We also corrected the statistical analysis.

- Figure 6B-D: It is not surprising that WT *var2csa*-expressing parasites also display VAR2CSA on their surface. For me, it would have been much more informative to analyze switcher and non-switcher lines with differences in transient *var2csa* activation (see comment above).

As mentioned above, we gather the reviewer is interested in whether the transient activation of *var2csa* transcription in the “switcher” lines results in any VAR2CSA protein on the RBC surface. We agree that this is an interesting question, and we now address this directly in the manuscript, as described in our response to a previous comment by the reviewer. In the experiment shown in Figure 6B, clone 1 is a switcher line and displays the early peak of *var2csa* expression, as shown in **Table EV5** (a non-switcher is also shown for comparison). Regardless of switcher or non-switcher status, we do not detect any binding to CSA in lines that do not stably express *var2csa*, indicating that the early peak of *var2csa* mRNA does not result in the expression of sufficient surface protein to mediate detectable binding. Similarly, we only detect surface expression of VAR2CSA when the transcript is stably expressed (Figure 6D), suggesting that the brief expression of the transcript early in the cycle does not lead to detectable amounts of protein on the RBC surface. This is perhaps not surprising given that the transient peak in *var2csa* transcripts in the switcher lines disappears before the PfEMP1 trafficking machinery is thought to be assembled and before RBC surface modification occurs. Thus, if any protein is made, it likely could not be exposed on the surface, at least not immediately.

- Figure 7: The legend indicates the panels A to C, but numbers are given in the figure. Please explain the abbreviation NMTGS (also only as an abbreviation in the main text, line 444).

We thank the reviewer for catching this error. We have corrected the figure accordingly.

- Line 560ff: How were the modified parasites selected? Subcloning?

At the reviewer’s request, we have updated the methods section to mention both drug selection and subcloning by limited dilution (see **lines 633-635**).

- Line 577: Typing, should be TRIzol, not TRiZol

We have made the correction as requested.

- Line 579ff: How was the cDNA synthesized? Using oligodT or random hexamers? Delete the space between "Super" and "Script", the correct name is SuperScript II.

We have added the requested details and corrections (see paragraph starting on **line 647**).

- The PCR mixture and cycle conditions for qPCR should be mentioned. Quantification using $2^{-\Delta\Delta Ct}$ requires not only equal amplification efficiency for normalizer and gene of interest ($=2$), but also for all var gene primers when comparing var gene X with var gene Y. Has this been verified experimentally?

The PCR conditions have been added to the Methods, as requested (see **lines 660-665**). This primer set was originally designed and experimentally validated for equal amplification efficiency by Salanti et al (2003), which we now mention in the text (see **line 657-658**). We routinely employ this primer set using genomic DNA as template and have similarly observed very similar amplification efficiencies.

- Line 604: Please mention the specificity (all var genes) and binding region of the primers used for exon 2 enrichment!

The complete list of primers is included in extended view **Table EV3**. We have added a description for how these oligos were designed (see **lines 687-690**).

- Line 605ff: Is the indicated Illumina kit, Illumina DNA Prep Kit, correct for RNAseq?

Yes. In consultation with engineers at Illumina, we first generated cDNA, then used the DNA Prep Kit.

- Line 608ff: The STAR argument basically excludes all reads with more than one mapping position, or in other words, only uniquely mapped reads were allowed. I would state this very clearly, as it increases the confidence that no exon 2 reads from other var (pseudo) genes are shown in Figure 3. Not var genes are multimapped, but their reads.

We added additional text to the methods as suggested by the reviewer (see **lines 697-698**). We also added a sentence to the main text to reinforce this point (see **lines 286-289**).

- Line 617ff: Please comment on the read length, the strain-specificity, the initial and down-sampled read depth.

We have updated the Methods section to specify that the read length was 150 bp and that the cDNA library used for sequencing was not strand-specific (see **lines 690-693**). We also added the range of reads obtained for the samples prior to down-sampling (see **lines 702-705**).

- Line 584ff: The formula for calculating relative expression assumes equal amplification efficiency for the normalizer and the gene of interest (=2) and does not produce values with a linear scale, e.g., $2^2 = 4$ and $2^{-2} = 0.25$ (requires log₁₀ transformation), Therefore, it is not valid to perform a t-test on the data, as shown in several figures.

We agree with the reviewer and have corrected the statistical analyses throughout the paper.

Any additional non-essential suggestions for improving the study (which will be at the author's/editor's discretion)

- In previous studies, the authors have shown subclonal lines with a so-called "many" phenotype. How do these parasites behave in terms of the expression of *var2csa* and exon 2 ncRNA, etc.?

We have performed preliminary experiments in a "many" line and observed a similar phenomenon regarding exon 2 expression from *var2csa*. However, given the complexity of the current manuscript, we prefer to leave this analysis to a future study.

- *Var2csa* exon 2 pseudogenes (clade A-ATS) terminate 5 of 7 internal *Var* gene clusters on the opposite DNA strand (Otto et al. 2018). Do you think they have a function within the switching network?

The reviewer makes a very interesting point here and we agree that these are potentially interesting genetic elements. We examined our RNAseq dataset to see if we could detect active transcription of these elements suggesting the ncRNAs might be made from them. However, we did not detect transcripts. We do not want to over-interpret these results since reads might be lost due to multimapping or because we did not specifically target these elements when designing unique primers for cDNA priming. Therefore, a thorough investigation of these elements will have to wait for future work.

Dear Kirk,

Thank you again for the submission of your revised manuscript (EMBOJ-2025-122057R) to The EMBO Journal for our consideration, and for your patience during re-review. As I have already informed you, the three original referees, who had also assessed the previous version of your manuscript, have now seen your revision, and we have received their comments, which are appended below.

I am pleased to say that all three referees are satisfied with the revision and recognize that their previous comments have been satisfactorily addressed. In light of this input, I am pleased to inform you that your manuscript has been accepted in principle for publication in The EMBO Journal.

There is only one remaining request (from referee #3) regarding the statistical test used for the analysis of the data presented in Figures 3C and 6A. The referee points out that a t-test is not adequate for the analysis of a small sample with data that are not normally distributed, and recommends either data log-transformation before analysis, or -preferably- the use of a non-parametric test. We kindly request that you address this remaining point completely in a final version of the manuscript, and also explain how it was addressed in a brief point-by-point response.

From the editorial side, there are also a few changes we need you to make in the final version of your manuscript, before we can move forward with its publication in The EMBO Journal:

- Please note that our notifications to the e-mail address vap4006@med.cornell.edu of your co-author Valay Patel could not be delivered. Please either remove this author from the list in the system and then add the name back using the current/new e-mail address, or send us the current/new e-mail address so that we can update the account on your co-author's behalf.
- Thank you for depositing the RNA sequencing, whole-genome sequencing, and CUT&RUN data to the NCBI sequence repository. We would be grateful if you could please include in the Data availability statement of the manuscript the accession IDs of these dataset(s).
- Please change heading "Disclosure of competing interests statement" to "Disclosure and competing interests statement".
- Please change heading "Materials and Methods" to "Methods".
- Please make sure that the funding information provided in the Acknowledgements section of the manuscript matches the information entered in our online manuscript handling system; please use the "More funders" option (not the Comments box) in the system to enter each funder/fellowship separately (Stavros S. Niarchos Scholar, William Randolph Hearst Endowed Faculty Fellowship, William Randolph Hearst Foundation).
- The 5 uploaded EV tables should be renamed to "Dataset EV1-Dataset EV5" and all files should be updated accordingly (i.e., source file names, titles in the system, legends, callouts in the main manuscript file and the reagents table); their legends need to be removed from the manuscript as they are already provided as a separate sheet in each Excel file.
- Our data editors have checked your Figures and their legends, and raised the following queries. Please address all points below completely in your revised manuscript (all changes should be highlighted or "tracked"):
 1. Please provide the exact p-values in the legend of Figure 6A.
 2. Please indicate the statistical test used for data analysis in the legend of Figure EV2.
 3. Please note that information related to "n" is missing in the legend of Figure EV2.
 4. Please note that the error bars must be defined in the legend of Figures 6B, D.
 5. Please note that the scale bar and its definition are missing in the legend of Figure 5G.
- Please note that EMBO press papers are accompanied online by:
 - A) a short (2 sentences) summary of the findings and their significance,
 - B) 2-5 short bullet points highlighting the key results, and
 - C) a synopsis image in .jpg or .png format that is exactly 550 pixels wide and 300-600 pixels high (the height is variable). Please note that all text needs to be legible at the final size.Please upload this information along with your revised manuscript (the text for A and B should be provided in a separate Word file).
- The manuscript sections need to be named and ordered as follows: Title page - Abstract - Introduction - Results - Discussion - Methods - Data Availability - Acknowledgements - Disclosure and Competing Interests Statement - References - Figure Legends - main Tables (if there are any) - Expanded View Figure Legends.
- Please also note that as part of the EMBO Press transparent editorial process, The EMBO Journal publishes online a Peer

Review File along with each accepted manuscript. This File will be published in conjunction with your paper and will include the referee reports, your point-by-point responses and all pertinent correspondence relating to the manuscript. Your Author's Checklist will also be published at the end of the Peer Review File. Please let us know in case you want to remove any data or figures from your point-by-point responses before they are published as part of the Peer Review File. Retaining unpublished data in the Peer Review File means that these count as published and that the Peer Review File would need to be referenced in future publications. Please let the editorial office know in case you want to remove any data from this file (contact@embojournal.org).

We look forward to seeing a final version of your manuscript as soon as possible. Please let us know if you have any questions and use this link to submit your revision: Link Unavailable.

Best regards,

Ioannis

Referee #1:

I have no more comments. The authors addressed my previous comments adequately. I recommend the manuscript for publication.

Referee #2:

The authors fully addressed my previous comments and I have no further questions.

Seb Baumgarten

Referee #3:

The authors have satisfactorily addressed all of my comments by providing additional data, discussion sections, references etc. However, the use of a t-test in Figures 3C and 6A remains inappropriate for the non-normally distributed data. Even with a Shapiro-Wilk test, the underlying data structure violates the assumptions of a t-test. The data should either be log-transformed prior to analysis or, preferably given the small sample size, analyzed using a non-parametric Mann-Whitney U test.

Dear Dr. Papaioannou,

Thank you again for handling our manuscript EMBOJ-2025-122057R for consideration by The EMBO Journal. We are very happy to hear that the paper has been accepted in principle.

Below please find a point-by-point description of the final modifications in response to the one comment from referee #3 and the additional comments from the editorial staff. We have included the original comment as well as our response (in blue text).

We hope the manuscript is now acceptable for publication in the EMBO Journal.

Thanks again for handling the manuscript. Please don't hesitate to contact me if any additional information is required.

Best regards,

Kirk Deitsch

There is only one remaining request (from referee #3) regarding the statistical test used for the analysis of the data presented in Figures 3C and 6A. The referee points out that a t-test is not adequate for the analysis of a small sample with data that are not normally distributed, and recommends either data log-transformation before analysis, or - preferably- the use of a non-parametric test. We kindly request that you address this remaining point completely in a final version of the manuscript, and also explain how it was addressed in a brief point-by-point response.

At the reviewer's request, we performed a non-parametric Mann-Whitney U test to analyze the data in Figures 3C and 6A. These statistical tests and their results are now shown in the figures and described in detail in the figure legends.

- Please note that our notifications to the e-mail address vap4006@med.cornell.edu of your co-author Valay Patel could not be delivered. Please either remove this author from the list in the system and then add the name back using the current/new e-mail address, or send us the current/new e-mail address so that we can update the account on your co-author's behalf.

As suggested, we have removed the author from the system then added his name back and included the new email address (valay_patel@fas.harvard.edu). Please let us know if this did not solve the problem.

- Thank you for depositing the RNA sequencing, whole-genome sequencing, and CUT&RUN data to the NCBI sequence repository. We would be grateful if you could please include in the Data availability statement of the manuscript the accession IDs of these dataset(s).

This has been added to the Data Availability section, as requested.

- Please change heading "Disclosure of competing interests statement" to "Disclosure and competing interests statement".

Corrected as requested.

- Please change heading "Materials and Methods" to "Methods".

Corrected as requested.

- Please make sure that the funding information provided in the Acknowledgements section of the manuscript matches the information entered in our online manuscript handling system; please use the "More funders" option (not the Comments box) in the system to enter each funder/fellowship separately (Stavros S. Niarchos Scholar, William Randolph Hearst Endowed Faculty Fellowship, William Randolph Hearst Foundation).

We have added the additional funders using the "More funders" option, as requested.

- The 5 uploaded EV tables should be renamed to "Dataset EV1-Dataset EV5" and all files should be updated accordingly (i.e., source file names, titles in the system, legends, callouts in the main manuscript file and the reagents table); their legends need to be removed from the manuscript as they are already provided as a separate sheet in each Excel file.

These have been corrected as requested.

- Our data editors have checked your Figures and their legends, and raised the following queries. Please address all points below completely in your revised manuscript (all changes should be highlighted or "tracked"):

1. Please provide the exact p-values in the legend of Figure 6A.

This has been added to the legend of Figure 6A.

2. Please indicate the statistical test used for data analysis in the legend of Figure EV2.

This has been added to the figure legend (DESeq2 (v.1.36) with a false discovery rate cutoff of < 0.05).

3. Please note that information related to "n" is missing in the legend of Figure EV2.

This has been added to the figure legend. The differential gene expression displayed in the volcano plots were derived from two independent replicate transcriptomes obtained for each population.

4. Please note that the error bars must be defined in the legend of Figures 6B, D. We have added definitions for the error bars in Figures 6B and D to the figure legend.

5. Please note that the scale bar and its definition are missing in the legend of Figure 5G.

The scale bar has been added to the image and its definition has been added to the figure legend.

- Please note that EMBO press papers are accompanied online by:

A) a short (2 sentences) summary of the findings and their significance,

B) 2-5 short bullet points highlighting the key results, and

C) a synopsis image in .jpg or .png format that is exactly 550 pixels wide and 300-600 pixels high (the height is variable). Please note that all text needs to be legible at the final size.

Please upload this information along with your revised manuscript (the text for A and B should be provided in a separate Word file).

These have been submitted as separate Word and jpg files along with the revised manuscript and figures.

- The manuscript sections need to be named and ordered as follows: Title page - Abstract - Introduction - Results - Discussion - Methods - Data Availability - Acknowledgements - Disclosure and Competing Interests Statement - References - Figure Legends - main Tables (if there are any) - Expanded View Figure Legends.

We have corrected the order of the sections as requested.

Dear Kirk,

Congratulations on an excellent manuscript! I am very pleased to inform you that it has been accepted for publication in The EMBO Journal. Thank you for comprehensively addressing the initially raised referee concerns and our editorial requests for corrections and other changes.

You may qualify for financial assistance for your publication charges - either via a Springer Nature fully open access agreement or an EMBO initiative. Check your eligibility: <https://link.springer.com/journal/44318/how-to-publish-with-us>

If you have any questions, please do not hesitate to contact the Editorial Office. Thank you for your contribution to The EMBO Journal. Working with you has been a pleasure.

Best regards,

Ioannis

Please note that it is The EMBO Journal policy for the transcript of the editorial process (containing referee reports and your response letters) to be published as an online supplement to each paper. If you should prefer removal of any referee-only figures included in the point-by-point response(s), e.g. because they may still be used for future publication or because they have been reproduced from published work by others, please do let us know immediately via response email.

More information is available here: <https://link.springer.com/partners/embo-press/editorial-policies#Peer%20review>